# THE COLIMIT OF METABELIEFS

## ABSTRACT

Potentially infinite sequences of beliefs arise when reasoning about the future, one's own beliefs, or others' beliefs. Machine learning researchers are typically content with heuristic truncation, or proofs of asymptotic convergence, of sequences of beliefs; however, such approaches lack insight into the structure of the possible choices. We construct and analyze several colimits of meta beliefs to understand the topological and geometric structure of sequences of beliefs. We analyze the relationship between different levels, the relationship between different beliefs at different levels, the encoding of temporal and other indexing structures in belief space, and structures preserved in the colimit. Examples demonstrate the ability to formalize and reason about problems of learning, cooperative and competitive communication, and sequential decision making. We conclude by emphasizing insights gained, and future directions for more concrete machine learning models.

Potentially infinite sequences of beliefs about beliefs—metabeliefs—arise when reasoning about the future, one's own, or other agents' beliefs. Reasoning about potential future observations gives rise to sequences of beliefs which quantify uncertainty about potential future states. One may have an infinite hierarchy of beliefs, each quantifying uncertainty at the previous level. Similarly, reasoning about other agents is a recursive process in which one's actions and/or beliefs are determined by considering what the other agents will think—an infinite sequence of beliefs.

In practice, metabeliefs are typically truncated to a finite sequence. In reinforcement learning, the introduction of a horizon or discount factor alleviates having to reason too far in the future (Sutton & Barto, 2018). In learning, hierarchies of beliefs are generally fixed and finite by assumption. When reasoning about other agents, Theory of Mind recursions are computed over a single estimate of the other agent's beliefs, without any acknowledgement of the uncertainty about unobservable properties of other agents (Camerer et al., 2004; Baker et al., 2011; 2017; Wang et al., 2020; Shum et al., 2019; Wright & Leyton-Brown, 2014).

However, a lack of rigorous analysis makes it difficult to understand the implications of practically-motivated assumptions. In particular, the relationship between the topology of the underlying space and how topological properties may be carried through to different levels of reasoning is often ignored. In some instances, there is a Wasserstein geometry which is induced at all levels of reasoning. In other cases, the interaction between an indexing set (say, discrete or continuous time) and sequences of evolving beliefs is ignored, though the topology of the indexing set itself impacts the structure of the possible beliefs about sequences of beliefs.

A concrete example is the game of rock-paper-scissors. In a single round of rock-paper-scissors, one must play one of the three options. Doing so depends on inferences about one's competitor, and beliefs about their choice are naturally represented as a probability distribution.[1] Because one can't know another's beliefs, it would be wise to consider *uncertainty* about those probabilities. Indeed, beliefs about beliefs are also uncertain, ad infinitum. Can we consider beliefs at all levels simultaneously?

We investigate the colimit of metabelief space as beliefs about beliefs about beliefs—ad infinitum—to understand the geometric and topological structure of metabeliefs. Section 1 defines and explores the colimit of beliefs: construction, topological properties, finite moment cases, evolving beliefs. Section 2 gives the relationship between the colimits constructed in Section 1. Section 3 gives several

---

[1] In contrast with Nash equilibria, our setting is applicable to non-cooperative games, cooperative games, and things that are not games. Here we focus in the rock-paper-scissors example on Bob not knowing Alice's strategy, her equilibrium strategy, or even exactly what her space of strategies might be.

examples and Section 4 an example computation. Section 5 gives related work. We conclude in Section 6 with future directions relating the colimit models to current machine-learning models. Throughout, we use rock-paper-scissors as a running example to illustrate the general framework. The proofs of all lemmas are contained in the Supplemental Materials.

# 1 BACKGROUND AND CONSTRUCTIONS

Let $\mathcal{X}$ be a metric space. Let $\mathcal{B}_{\mathcal{X}}$ denote the Borel $\sigma$-algebra with respect to the topology induced by the metric on $\mathcal{X}$. Consider the space $\mathcal{P}(\mathcal{X})$ of Borel probability measures on $\mathcal{X}$. We can topologize $\mathcal{P}(\mathcal{X})$ by defining a base of open neighborhoods for any $\mu \in \mathcal{P}(\mathcal{X})$ as follows: Consider the family of sets defined via the following,

$$V_\mu(f_1, ..., f_k; \epsilon_1, ..., \epsilon_k) = \left\{ \nu \in \mathcal{P}(\mathcal{X}) \text{ s.t. } \left| \int_{\mathcal{X}} f_i d\mu - \int_{\mathcal{X}} f_i d\nu \right| < \epsilon_i, i = 1, ..., k \right\}. \quad (1)$$

In the above, $f_1, ..., f_k$ are continuous functions on $\mathcal{X}$ and $\epsilon_1, ..., \epsilon_k > 0$. The topology generated by the collection of all such sets, varying $k$, $f_1, ..., f_k$, and $\epsilon_1, ..., \epsilon_k$ for every $\mu \in \mathcal{P}(\mathcal{X})$ defines the weak topology on $\mathcal{P}(\mathcal{X})$.

**Theorem 1.** (*Parthasarathy, 2005*) *Consider the weak topology on $\mathcal{P}(\mathcal{X})$ defined above.*

1. *Suppose that $\mathcal{X}$ is a metrizable space consisting of at least two points. Then $\mathcal{X}$ is homeomorphic to the subspace $\{\delta_x | x \in \mathcal{X}\}$ in $\mathcal{P}(\mathcal{X})$ consisting of single-atom Dirac distributions.*
2. *$\mathcal{P}(\mathcal{X})$ can be metrized as a separable metric space if and only if $\mathcal{X}$ is a separable metric space.*
3. *Suppose that $\mathcal{X}$ is a separable metric space and $\mathcal{E} \subseteq \mathcal{X}$ is dense in $\mathcal{X}$. Then the set of all measures whose supports are finite subsets of $\mathcal{E}$ is dense in $\mathcal{P}(\mathcal{X})$.*
4. *$\mathcal{P}(\mathcal{X})$ is a (para)compact metric space if and only if $\mathcal{X}$ is a (para)compact metric space.*
5. *Suppose that $\mathcal{X}$ is a separable metric space. Then $\mathcal{P}(\mathcal{X})$ is homeomorphic to a complete metric space if and only if $\mathcal{X}$ is homeomorphic to a complete metric space.*

Let $\mathcal{P}^{n+1}(\mathcal{X}) := \mathcal{P}(\mathcal{P}^n(\mathcal{X}))$ for $n \geq 0$, and $\mathcal{P}^0(\mathcal{X}) := \mathcal{X}$. Furthermore, let $\delta^n : \mathcal{P}^n(\mathcal{X}) \to \mathcal{P}^{n+1}(\mathcal{X})$ be the map sending $p \in \mathcal{P}^n(\mathcal{X})$ to the single-atom Dirac distribution centered at $p$. The first part of Theorem 1 indicates that $\mathcal{X}$ is homeomorphic to the subspace $\{\delta^n \circ \cdots \circ \delta^0(x) | x \in \mathcal{X}\}$ in $\mathcal{P}^{n+1}(\mathcal{X})$, since a composition of homeomorphisms is a homeomorphism. Parts (2)-(5) of Theorem 1 indicate how the topology and metrizability of each $\mathcal{P}^n(\mathcal{X})$ is influenced by the topology and metrizability of the initial space $\mathcal{X}$.

We wish to consider some way of taking the limit as $n \to \infty$ of $\mathcal{P}^n(\mathcal{X})$ in order to study all levels of reasoning simultaneously; one way of doing this is to compute the direct limit (or colimit) of the following sequence:

$$\mathcal{X} \xrightarrow{\delta^0} \mathcal{P}(\mathcal{X}) \xrightarrow{\delta^1} \mathcal{P}^2(\mathcal{X}) \xrightarrow{\delta^2} \cdots . \quad (2)$$

Doing so will ensure that information about how the distributions at a lower level, say in $\mathcal{P}^n(\mathcal{X})$ for $n \geq 0$ become Dirac distributions at higher levels, i.e. in any $\mathcal{P}^{n+m}(\mathcal{X})$ for any $m \geq 1$. In order to compute the direct limit of this sequence, we must first define a *direct system*[2] over the indexing set $\mathbb{N}$. To this end we define the following functions, $\delta^{ij} : \mathcal{P}^i(\mathcal{X}) \to \mathcal{P}^j(\mathcal{X})$ for $i \leq j$, $i, j \in \mathbb{N}$:

1. $\delta^{ii} : \mathcal{P}^i(\mathcal{X}) \to \mathcal{P}^i(\mathcal{X})$ is the identity map on $\mathcal{P}^i(\mathcal{X})$.
2. $\delta^{ij} : \mathcal{P}^i(\mathcal{X}) \to \mathcal{P}^j(\mathcal{X})$ for $i < j$ is defined as $\delta^{ij} = \delta^{j-1} \circ \cdots \circ \delta^i$.

Note that in particular for any $i \leq j \leq k$, $\delta^{ik} = \delta^{jk} \circ \delta^{ij}$. We now wish to construct the direct limit of this direct system $\langle \mathcal{P}^i(\mathcal{X}), \delta^{ij} \rangle$. This is a space, which we will denote suggestively by $\mathcal{P}^\infty(\mathcal{X})$, together with maps $\Delta^i : \mathcal{P}^i(\mathcal{X}) \to \mathcal{P}^\infty(\mathcal{X})$, such that for any topological space $\mathcal{Y}$ with continuous maps $\psi_i : \mathcal{P}^i(\mathcal{X}) \to \mathcal{Y}$ there exists a unique continuous function $u : \mathcal{P}^\infty(\mathcal{X}) \to \mathcal{Y}$ such that the

---

[2]The definition of a direct system is given by the sequence of topological spaces together with the maps $\delta^{ij}$ between them, satisfying conditions (1) and (2).

following diagram commutes:

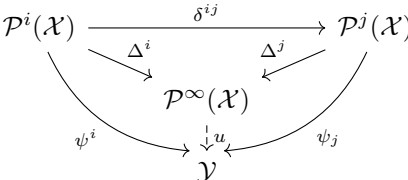

That is, for every $\mu \in \mathcal{P}^i(\mathcal{X})$, we have that $\Delta^j \circ \delta^{ij}(\mu) = \Delta^i(\mu)$, $\psi^i(\mu) = u \circ \Delta^i(\mu)$, and for any $\nu \in \mathcal{P}^j(\mathcal{X})$, $\psi_j(\nu) = u \circ \Delta^j(\nu)$. This means that continuous maps "factor through" the direct limit space $\mathcal{P}^\infty(\mathcal{X})$, so it encodes information about, for example, continuously sampling from distributions at all levels of reasoning.

## 1.1 CONSTRUCTING $\mathcal{P}^\infty(\mathcal{X})$

We will construct $\mathcal{P}^\infty(\mathcal{X})$ in two pieces: first, we will define the underlying set of $\mathcal{P}^\infty(\mathcal{X})$; then, we will define the topology on $\mathcal{P}^\infty(\mathcal{X})$, known as the 'final topology.' This is a standard construction from the theory of colimits for topological spaces, see e.g. (Mac Lane, 2013). To construct the underlying set, we first consider the following set:

$$\mathcal{S} := \bigsqcup_{n \in \mathbb{N}} \mathcal{P}^n(\mathcal{X}). \tag{3}$$

Now we define an equivalence relation $\sim$ on $\mathcal{S}$ by saying that for any $\mu \in \mathcal{P}^i(\mathcal{X})$ and $\nu \in \mathcal{P}^j(\mathcal{X})$, $\mu \sim \nu$ ($\mu$ is equivalent to $\nu$) if and only if there is some $k \in \mathbb{N}$ such that $i \leq k$ and $j \leq k$ with $\delta^{ik}(\mu) = \delta^{jk}(\nu)$. Then, as a set, $\mathcal{P}^\infty(\mathcal{X})$ is equal to the set of equivalence classes in $\mathcal{S}$. That is, we have the following:

$$\mathcal{P}^\infty(\mathcal{X}) = \left( \bigsqcup_{n \in \mathbb{N}} \mathcal{P}^n(\mathcal{X}) \right) \Big/ \sim. \tag{4}$$

We may then define maps $\Delta^i : \mathcal{P}^i(\mathcal{X}) \to \mathcal{P}^\infty(\mathcal{X})$ which send each distribution to its corresponding equivalence class.

**Lemma 2.** *Every equivalence class in $\mathcal{P}^\infty(\mathcal{X})$ is of the form $\{\mu, \delta^n(\mu), \delta^{n+1} \circ \delta^n(\mu), \delta^{n+2} \circ \delta^{n+1} \circ \delta^n(\mu), ...\}$ where $\mu$ is a unique distribution in $\mathcal{P}^n(\mathcal{X})$ such that $\mu$ is not a single-atom Dirac distribution or $\mu$ is a unique point in $\mathcal{X}$ (recall that $\delta^0(x) = \delta_x$ is the Dirac distribution in $\mathcal{P}(\mathcal{X})$ with support equal to $\{x\}$).*

The topology on $\mathcal{P}^\infty(\mathcal{X})$ is what is known as the *final topology*, which consists of the topology with the maximum collection of open sets such that each $\Delta^i : \mathcal{P}^i(\mathcal{X}) \to \mathcal{P}^\infty(\mathcal{X})$ is continuous.

## 1.2 PROPERTIES OF $\mathcal{P}^\infty(\mathcal{X})$

In order to better understand the structure of the colimit $\mathcal{P}^\infty(\mathcal{X})$, we collect here some properties of the space. For example, what properties of $\mathcal{X}$ filter through to $\mathcal{P}^\infty(\mathcal{X})$? We know from Theorem 1, for example, that if $\mathcal{X}$ is compact, then so is $\mathcal{P}^n(\mathcal{X})$ for every $n \in \mathbb{N}$. However, compactness is not translated to the colimit.

**Lemma 3.** *Suppose that $\mathcal{X}$ is a compact T1 space[3] consisting of at least two distinct points. Then $\mathcal{P}^\infty(\mathcal{X})$ is non-compact with respect to the final topology.*

In the following theorem, we will show that some topological properties do pass to the colimit.

**Theorem 4.** *Suppose that $\mathcal{X}$ is a metrizable paracompact Hausdorff space. Then $\mathcal{P}^\infty(\mathcal{X})$ is a metrizable paracompact Hausdorff space.*

In order to prove the above theorem, it will be useful to first show that the property of being Hausdorff translates from $\mathcal{X}$ to $\mathcal{P}^n(\mathcal{X})$ for every $n \in \mathbb{N}$.

**Lemma 5.** *If $\mathcal{X}$ is Hausdorff, then so is $\mathcal{P}(\mathcal{X})$.*

---

[3]This is equivalent to saying that for every $x \in \mathcal{X}$, the set $\{x\}$ is a closed subset of $\mathcal{X}$.

*Proof.* (Proof of Theorem 4) By properties (1) and (4) of Theorem 1, if $\mathcal{X}$ is metrizable then so is $\mathcal{P}^n(\mathcal{X})$ and if $\mathcal{X}$ is paracompact, then so is $\mathcal{P}^n(\mathcal{X})$ for every $n \in \mathbb{N}$. Furthermore, if $\mathcal{X}$ is Hausdorff, then so is $\mathcal{P}^n(\mathcal{X})$ for every $n \in \mathbb{N}$ by Lemma 5. Therefore the result follows from (Michael, 1956) □

**Example: rock-paper-scissors.** Suppose that two agents, Alice and Bob, are playing rock-paper-scissors. Let $\mathcal{X} = \{\text{rock-paper-scissors}\}$ be the space of Alice's potential moves. In order for Bob to generate his own strategy, he must consider what he knows (or believes that he knows) about Alice, which results in a distribution in $\mathcal{P}(\mathcal{X})$ where Bob assigns a certain probability to choosing each potential move. However, a sophisticated Bob must also contend with his uncertainty about his own beliefs, i.e. he may consider a distribution in $\mathcal{P}(\mathcal{P}(\mathcal{X})) = \mathcal{P}^2(\mathcal{X})$, and sampling from this distribution will give him a belief about Alice's potential move. Similarly, an even more sophisticated Bob will consider how uncertain he is about his own uncertainty, and be operating in $\mathcal{P}^4(\mathcal{X})$. As the player Bob becomes more sophisticated, he will be operating in $\mathcal{P}^n(\mathcal{X})$ for increasingly large values of $n$. The colimit space $\mathcal{P}^\infty(\mathcal{X})$ contains information about all possible Bobs, regardless of sophistication. If Alice is uncertain about what sort of player Bob is, in reasoning about how Bob is reasoning about her strategy, Alice is (implicitly) operating in the colimit space $\mathcal{P}^\infty(\mathcal{X})$.

## 1.3 CONSTRUCTING $\mathcal{P}_p^\infty(\mathcal{X})$

As Theorem 4 shows, if $\mathcal{X}$ is a metrizable paracompact Hausdorff space, so is $\mathcal{P}^\infty(\mathcal{X})$. In this section we will consider $\mathcal{X}$ to be a separable complete metric space with metric $d : \mathcal{X} \times \mathcal{X} \to \mathbb{R}_{\geq 0}$, and study how the metric structure gives rise to metric structures on a subspace of $\mathcal{P}^n(\mathcal{X})$ for every $n \in \mathbb{N}$, as well as on the colimit of these subspaces.

Let $p \geq 1$ and consider the space $\mathcal{P}_p(\mathcal{X}) = \{\mu \in \mathcal{P}(\mathcal{X}) | \int_{\mathcal{X}} d(x,y)^p d\mu(y) < \infty \; \mu - a.e. \; x \in \mathcal{X}\}$ of distributions with finite $p$-th moments almost everywhere. Then we may construct a similar direct system:

$$\mathcal{X} \xrightarrow{\delta^0} \mathcal{P}_p(\mathcal{X}) \xrightarrow{\delta^1} \mathcal{P}_p^2(\mathcal{X}) \xrightarrow{\delta^2} \cdots . \tag{5}$$

We could generalize this example by changing the values of $p$ at each level, but the notation quickly becomes unwieldy for something that is likely not of much interest. At any rate, the sequence in Equation 5 may be turned into a direct system as before, and we may again construct the colimit $\mathcal{P}_p(\mathcal{X})$. Similarly to Lemma 2, each equivalence class has a unique non-single-atom Dirac distribution. In this case, since $\mathcal{X}$ is separable and complete, so is $\mathcal{P}_p^n(\mathcal{X})$ for any finite $n$, with respect to the Wasserstein $p$-metric (Bogachev & Kolesnikov, 2012). Furthermore, the topology induced by the Wasserstein $p$-metric coincides with the weak topology defined previously. We will denote the Wasserstein $p$-metric on $\mathcal{P}_p^n(\mathcal{X})$ by $W_p^n$. That is, when $n = 0$, $W_p^0(x,y) = d(x,y)$, and when $n \geq 1$ and for $\mu, \nu \in \mathcal{P}_p^n(\mathcal{X})$, we have:

$$W_p^n(\mu, \nu) := \left( \inf_{\gamma \in \Gamma(\mu,\nu)} \mathbb{E}_{(\alpha,\beta) \sim \gamma} \left[ \left( W_p^{n-1}(\alpha, \beta) \right)^p \right] \right)^{\frac{1}{p}} . \tag{6}$$

In the above, $\Gamma(\mu, \nu) \subseteq \mathcal{P}\left(\mathcal{P}_p^{n-1}(\mathcal{X}) \times \mathcal{P}_p^{n-1}(\mathcal{X})\right)$ is the space of joint distributions on $\mathcal{P}_p^{n-1}(\mathcal{X}) \times \mathcal{P}_p^{n-1}(\mathcal{X})$ having marginals $\mu$ and $\nu$.

In order to define a Wasserstein-like metric on $\mathcal{P}_p^\infty(\mathcal{X})$, we will first make a few definitions for ease of writing. For $[\mu] \in \mathcal{P}_p^\infty(\mathcal{X})$ an equivalence class, define the *rank* of $[\mu]$, denoted $r_{[\mu]}$, as the unique $n \in \mathbb{N}$ such that there exists $\hat{\mu} \in [\mu] \cap \mathcal{P}_p^n(\mathcal{X})$ where $\hat{\mu}$ is not a single-atom Dirac distribution. For ease of notation, we will also denote $\max\{r_{[\mu]}, r_{[\nu]}\}$ as $m([\mu], [\nu])$ for two equivalence classes $[\mu], [\nu] \in \mathcal{P}_p^\infty(\mathcal{X})$. Now, we define a metric on $\mathcal{P}_p^\infty(\mathcal{X})$ as follows:

$$W_p^\infty([\mu], [\nu]) := W_p^{m([\mu],[\nu])}(\mu_{m([\mu],[\nu])}, \nu_{m([\mu],[\nu])}). \tag{7}$$

Here $\mu_{m([\mu],[\nu])}$ (resp. $\nu_{m([\mu],[\nu])}$) denotes the unique element in $[\mu] \cap \mathcal{P}_p^{m([\mu],[\nu])}$ (resp. $[\nu] \cap \mathcal{P}_p^{m([\mu],[\nu])}$). That is, one takes representatives from $[\mu], [\nu]$ in a space $\mathcal{P}_p^n(\mathcal{X})$ such that at least one of the representatives is a non-single-atom Dirac distribution, and computes the Wasserstein $p$-distance between these two representatives.[4]

---

[4]Note that from the properties of the Wasserstein distance, it is sufficient to consider in Equation 7 $W_p^m$ for any $m \geq m([\mu], [\nu])$.

**Lemma 6.** $W_p^\infty$ *is a metric on* $\mathcal{P}_p^\infty(\mathcal{X})$.

**Theorem 7.** *The metric topology on* $\mathcal{P}_p^\infty(\mathcal{X})$ *induced by* $W_p^\infty$ *coincides with the final topology.*

*Proof.* Suppose $E \subseteq \mathcal{P}_p^\infty(\mathcal{X})$ is open with respect to the metric topology. That is, for some indexing set $A$ there exist open balls $B_{\epsilon_{t_a}}^\infty([\mu_{t_a}])$ such that we may write $E$ as follows:

$$E = \bigcup_{a \in A} \bigcap_{t_a=1}^{n_a} B_{\epsilon_{t_a}}^\infty([\mu_{t_a}]). \tag{8}$$

For convenience, we will suppose that the representative $\mu_{t_a} \in [\mu_{t_a}]$ is such that $\mu_{t_a} \in \mathcal{P}_p^{r_{[\mu_{t_a}]}}(\mathcal{X})$, i.e. $\mu_{t_a}$ is the unique non-single-atom-Dirac-distribution element of $[\mu_{t_a}]$. We would like to show that $E$ is open with respect to the final topology. It is necessary and sufficient to prove that for each $i \in \mathbb{N}$, $(\Delta^i)^{-1}(E)$ is open in $\mathcal{P}_p^i(\mathcal{X})$. We compute:

$$(\Delta^i)^{-1}(E) = \{\nu \in \mathcal{P}_p^i(\mathcal{X}) | \Delta^i(\nu) \in E\} \tag{9}$$

$$= \bigcup_{a \in A} \bigcap_{t_a=1}^{n_a} \left\{ \nu \in \mathcal{P}_p^i(\mathcal{X}) \middle| W_p^{m(i, r_{[\mu_{t_a}]})}(\hat{\nu}, \hat{\mu}_{t_a}) < \epsilon_{t_a} \right\}.$$

Here $\hat{\nu}, \hat{\mu}_{t_a}$ are representatives in $\mathcal{P}_p^{m(i, r_{[\mu_{t_a}]})}(\mathcal{X})$ of the equivalence classes of $[\nu], [\mu_{t_a}]$, respectively. Equation 9 then becomes:

$$\bigcup_{a \in A} \bigcap_{t_a=1}^{n_a} \left\{ \nu \in \mathcal{P}_p^i(\mathcal{X}) \middle| \begin{cases} W_p^i(\nu, \delta^{r_{[\mu_{t_a}]}i}(\mu_{t_a})) < \epsilon_{t_a} & \text{if } i \geq r_{[\mu_{t_a}]} \\ W_p^{r_{[\mu_{t_a}]}}(\delta^{ir_{[\mu_{t_a}]}}(\nu), \mu_{t_a}) < \epsilon_{t_a} & \text{if } i < r_{[\mu_{t_a}]} \end{cases} \right\} \tag{10}$$

$$= \bigcup_{a \in A} \bigcap_{t_a=1}^{n_a} \begin{cases} B_{\epsilon_{t_a}}^i \left( \delta^{r_{[\mu_{t_a}]}i}(\mu_{t_a}) \right) & \text{if } i \geq r_{[\mu_{t_a}]} \\ (\delta^{ir_{[\mu_{t_a}]}})^{-1} \left( \delta^{ir_{[\mu_{t_a}]}}(\mathcal{P}_p^i(\mathcal{X})) \cap B_{\epsilon_{t_a}}^{r_{[\mu_{t_a}]}}(\mu_{t_a}) \right) & \text{if } i < r_{[\mu_{t_a}]} \end{cases}.$$

Note that in the case that $i \geq r_{[\mu_{t_a}]}$, we have open balls in $\mathcal{P}_p^\infty(\mathcal{X})$. Moreover, in the case $i < r_{[\mu_{t_a}]}$, we have an open set since $\delta^{ir_{[\mu_{t_a}]}}(\mathcal{P}_p^i(\mathcal{X})) \cap B_{\epsilon_{t_a}}^{r_{[\mu_{t_a}]}}(\mu_{t_a})$ is an open set with respect to the subspace topology on $\delta^{ir_{[\mu_{t_a}]}}(\mathcal{P}_p^i(\mathcal{X})) \subseteq \mathcal{P}_p^{r_{[\mu_{t_a}]}}(\mathcal{X})$, and $\delta^{ir_{[\mu_{t_a}]}}$ is a homeomorphism onto its image, so necessarily continuous. Thus, $(\Delta^i)^{-1}(E)$ is open in $\mathcal{P}_p^i(\mathcal{X})$ for every $i \in \mathbb{N}$, so $E$ is open with respect to the final topology.

Conversely, suppose that we have an open set $E$ with respect to the final topology. We now wish to find a decomposition as in Equation 8 to show that $E$ is open with respect to the metric topology. Because $E$ is open with respect to the final topology on $\mathcal{P}_p^\infty(\mathcal{X})$, $(\Delta^i)^{-1}(E) \subseteq \mathcal{P}_p^i(\mathcal{X})$ is open. Since the weak topology on $\mathcal{P}_p^i(\mathcal{X})$ coincides with the metric topology induced by the Wasserstein $p$-metric $W_p^i$, there must exist the following decomposition for each $i \in \mathbb{N}$:

$$(\Delta^i)^{-1}(E) = \bigcup_{a \in A_i} \bigcap_{t_a^i=1}^{n_a^i} B_{\epsilon_{t_a^i}}^i(\nu_{t_a^i}). \tag{11}$$

Applying $\Delta^i$ to both sides yields:

$$\{[\nu] \in E | r_{[\nu]} \leq i\} = \bigcup_{a \in A_i} \Delta^i \left( \bigcap_{t_a^i=1}^{n_a^i} B_{\epsilon_{t_a^i}}^i(\nu_{t_a^i}) \right) \tag{12}$$

Therefore, $E$ may be written as:

$$E = \bigcup_{i \in \mathbb{N}} \bigcup_{a \in A_i} \Delta^i \left( \bigcap_{t_a^i=1}^{n_a^i} B_{\epsilon_{t_a^i}}^i(\nu_{t_a^i}) \right) \tag{13}$$

**Claim:** For two sets $B_1, B_2 \subseteq \mathcal{P}^i(\mathcal{X})$, we have $\Delta^i(B_1 \cap B_2) = \Delta^i(B_1) \cap \Delta^i(B_2)$.

*Proof of claim:* Let $[\nu] \in \Delta^i(B_1 \cap B_2)$ such that $\nu$ is the unique representative of $[\nu]$ that is not a single-atom Dirac distribution. Then there exists $\mu \in B_1 \cap B_2$ such that $\Delta^i(\mu) = [\nu]$, so $[\nu] \in \Delta^i(B_1) \cap \Delta^i(B_2)$. Conversely, suppose that $[\nu] \in \Delta^i(B_1) \cap \Delta^i(B_2)$, again with $\nu \in \mathcal{P}_p^k(\mathcal{X})$ the unique non-single atom Dirac distribution in $[\nu]$. Note that $k \leq i$. Then there exist $\nu_j \in B_j$,

$j = 1, 2$, such that $\Delta^i(\nu_j) = [\nu]$. By definition of $\Delta^i$, this implies $\nu_j = \delta^{ki}(\nu)$, hence $\nu_1 = \nu_2$, so $\nu_1 \in B_1 \cap B_2$. Thus $[\nu] \in \Delta^i(B_1 \cap B_2)$.

Now, Equation 13 becomes:

$$E = \bigcup_{i \in \mathbb{N}} \bigcup_{a \in A_i} \bigcap_{t_a^i = 1}^{n_a^i} \Delta^i \left( B^i_{\epsilon_{t_a^i}} \left( \nu_{t_a^i} \right) \right) \tag{14}$$

$$= \bigcup_{i \in \mathbb{N}} \bigcup_{a \in A_i} \bigcap_{t_a^i = 1}^{n_a^i} \left( B^\infty_{\epsilon_{t_a^i}} \left( [\nu_{t_a^i}] \right) \cap \Delta^i \left( \mathcal{P}^i(\mathcal{X}) \right) \right) = \bigcup_{i \in \mathbb{N}} \bigcup_{a \in A_i} \left( \Delta^i \left( \mathcal{P}^i(\mathcal{X}) \right) \cap \left( \bigcap_{t_a^i = 1}^{n_a^i} B^\infty_{\epsilon_{t_a^i}} \left( [\nu_{t_a^i}] \right) \right) \right)$$

$$= \bigcup_{a \in \cup_{i \in \mathbb{N}} A_i} \bigcap_{t_a = 1}^{n_a} B^\infty_{\epsilon_{t_a}} \left( [\nu_{t_a}] \right).$$

In the above we have reindexed by combining all of the indexing sets $A_i$ into one. Therefore, $E$ is open with respect to the metric topology on $\mathcal{P}_p^\infty(\mathcal{X})$. $\square$

## 1.4 Index-dependent distributions

Suppose that we wish to include the beliefs evolving with respect to some parameter. For example, the parameter could be discrete time (indexed by $\mathbb{N}$) or continuous time (indexed by $\mathbb{R}$) or some more abstract quantity. To this end, the space of evolving beliefs becomes the space of continuous maps from the indexing space, $\mathcal{I}$, to the space of probability distributions on the underlying space, $\mathcal{P}(\mathcal{X})$. We denote this space by $Hom(\mathcal{I}, \mathcal{P}(\mathcal{X}))$. Higher levels of reasoning which also evolve as indexed by $\mathcal{I}$ may then be quantified via letting $\mathcal{H}_{\mathcal{I}}\mathcal{P}^{n+1}(\mathcal{X}) := Hom(\mathcal{I}, \mathcal{P}(\mathcal{H}_{\mathcal{I}}\mathcal{P}^n(\mathcal{X})))$, with $\mathcal{H}_{\mathcal{I}}\mathcal{P}^0(\mathcal{X}) := \mathcal{X}$. Now, we define the topology on $\mathcal{H}_{\mathcal{I}}\mathcal{P}(\mathcal{X})$ to be the *compact-open topology*. For any compact subset[5] $K \subseteq \mathcal{I}$ and open set[6] $U \subseteq \mathcal{P}(\mathcal{X})$, we define:

$$V(K, U) := \{ f \in \mathcal{H}_{\mathcal{I}}\mathcal{P}(\mathcal{X}) | f(K) \subseteq U \}. \tag{15}$$

The collection of all such $V(K, U)$ then forms a sub-base for the compact-open topology on $\mathcal{H}_{\mathcal{I}}\mathcal{P}(\mathcal{X})$.

**Lemma 8.** *Any space $\mathcal{X}$ may be embedded continuously in $\mathcal{H}_{\mathcal{I}}\mathcal{P}(\mathcal{X})$ via the following map, for any $x \in \mathcal{X}$ and $i \in \mathcal{I}$:*

$$cd(x)(i) := \delta_x. \tag{16}$$

*That is, $cd$ is the composition of the map sending each point $x$ to the single-atom Dirac distribution centered at $x$ and the constant-valued map sending each point in $\mathcal{I}$ to the aforementioned distribution.*

Similarly to the previous, index-independent case, we may then define maps $(cd)^n : \mathcal{P}^n(\mathcal{X}) \to \mathcal{P}^{n+1}(\mathcal{X})$ with $(cd)^n(p)(i) = \delta_p$.

**Lemma 9.** *Suppose that $\mathcal{X}$ is Hausdorff. Then each map $(cd)^n$ defined above is a homeomorphism onto its image.*

Again, we may also define the maps $(cd)^{ij} : \mathcal{P}^i(\mathcal{X}) \to \mathcal{P}^j(\mathcal{X})$ for $i \leq j$ by taking $(cd)^{ii}$ to be the identity map on $\mathcal{P}^i(\mathcal{X})$ and for $j > i$, $(cd)^{ij} = (cd)^{j-1} \circ \cdots \circ (cd)^i$ is a map embedding $\mathcal{P}^i(\mathcal{X})$ into $\mathcal{P}^j(\mathcal{X})$. Therefore, we have a direct system, and so the colimit $\mathcal{P}^\infty(\mathcal{X})$ exists.

Now, we may relate any space $Hom(\mathcal{I}, A)$ to $\mathcal{P}(A)$ by defining a map called *push* from $\mathcal{P}(\mathcal{I}) \times Hom(\mathcal{I}, A) \to \mathcal{P}(A)$ as follows: For any $\mu \in \mathcal{P}(\mathcal{I})$ and $f \in Hom(\mathcal{I}, A)$, and any measurable $E \subseteq A$, we have:

$$push(\mu, f) := f^*(\mu)(E) = \mu(f^{-1}(E)). \tag{17}$$

That is, $push(\mu, f)$ is the push-forward of $\mu$ along $f$. In particular, we obtain a map $push : H_{\mathcal{I}}\mathcal{P}^{n+1}(\mathcal{X}) \to \mathcal{P}^2(\mathcal{H}_{\mathcal{I}}\mathcal{P}^n(\mathcal{X}))$.

**Lemma 10.** *The following diagram commutes:*

$$
\begin{array}{ccc}
\mathcal{P}(\mathcal{I}) \times \mathcal{H}_{\mathcal{I}}\mathcal{P}^n(\mathcal{X}) & \xrightarrow{id \times \delta^0} & \mathcal{P}(\mathcal{I}) \times \mathcal{P}(\mathcal{H}_{\mathcal{I}}\mathcal{P}^n(\mathcal{X})) \\
{\scriptstyle id \times (cd)^n} \downarrow & & \downarrow {\scriptstyle \delta^1 \circ \pi_2} \\
\mathcal{P}(\mathcal{I}) \times \mathcal{H}_{\mathcal{I}}\mathcal{P}^{n+1}(\mathcal{X}) & \xrightarrow{push} & \mathcal{P}^2(\mathcal{H}_{\mathcal{I}}\mathcal{P}^n(\mathcal{X}))
\end{array}
$$

---

[5] Note that $\mathcal{I}$ should necessarily come with its own topology, and hence compact subsets.

[6] Here we are using the weak topology on $\mathcal{P}(\mathcal{X})$ as defined previously.

Here $\pi_2 : \mathcal{P}(\mathcal{I}) \times \mathcal{P}(\mathcal{H}_\mathcal{I}\mathcal{P}^n(\mathcal{X})) \to \mathcal{P}(\mathcal{H}_\mathcal{I}\mathcal{P}^n(\mathcal{X}))$ is projection onto the second coordinate; i.e. $\pi_2(\mu, \nu) = \nu$.

In particular, this shows that the map $push \circ (id \times (cd)^n) : \mathcal{P}(\mathcal{I}) \times \mathcal{H}_\mathcal{I}\mathcal{P}^n(\mathcal{X}) \to \mathcal{P}^2(\mathcal{H}_\mathcal{I}\mathcal{P}^n(\mathcal{X}))$ is independent of the choice of distribution $\mu \in \mathcal{I}$.

Similar to the diagram in Lemma 10, we may also consider evaluating at a point in $\mathcal{I}$ in order to get a distribution on $\mathcal{H}_\mathcal{I}\mathcal{P}^n(\mathcal{X})$.

**Lemma 11.** *The following diagram commutes:*

$$
\begin{array}{ccc}
\mathcal{I} \times \mathcal{H}_\mathcal{I}\mathcal{P}^n(\mathcal{X}) & \xrightarrow{\pi_2} & \mathcal{H}_\mathcal{I}\mathcal{P}^n(\mathcal{X}) \\
{\scriptstyle id \times (cd)^n} \downarrow & & \downarrow {\scriptstyle \delta} \\
\mathcal{I} \times \mathcal{H}_\mathcal{I}\mathcal{P}^{n+1}(\mathcal{X}) & \xrightarrow{ev} & \mathcal{P}(\mathcal{H}_\mathcal{I}\mathcal{P}^n(\mathcal{X})).
\end{array}
$$

*Here $ev : \mathcal{I} \times \mathcal{H}_\mathcal{I}\mathcal{P}^{n+1}(\mathcal{X}) \to \mathcal{P}(\mathcal{H}_\mathcal{I}\mathcal{P}^n(\mathcal{X}))$ is the evaluation map satisfying $ev(i, f) = f(i)$.*

## 2 RELATIONSHIPS BETWEEN THE COLIMITS

We now study how the different colimit spaces $\mathcal{P}^\infty(\mathcal{X})$, $\mathcal{P}_p^\infty(\mathcal{X})$, and $\mathcal{H}_\mathcal{I}\mathcal{P}^\infty(\mathcal{X})$ are related. In particular, we will show that $\mathcal{P}_p^\infty(\mathcal{X})$ is embedded into $\mathcal{P}^\infty(\mathcal{X})$, that $\mathcal{P}^\infty(\mathcal{X})$ may be embedded into $\mathcal{H}_\mathcal{I}\mathcal{P}^\infty(\mathcal{X})$, and how we can pass from $\mathcal{I}^\infty \times \mathcal{H}_\mathcal{I}\mathcal{P}^\infty(\mathcal{X})$ back to $\mathcal{P}^\infty(\mathcal{X})$.

Since $\mathcal{P}_p^n(\mathcal{X})$ is a subset of $\mathcal{P}^n(\mathcal{X})$ for every $n \in \mathbb{N}$, we have that there is an induced map $\iota : \mathcal{P}_p^\infty(\mathcal{X}) \to \mathcal{P}^\infty(\mathcal{X})$ which sends each equivalence class $[\mu] \in \mathcal{P}_p^\infty(\mathcal{X})$ to the equivalence class $[\mu] \in \mathcal{P}^\infty(\mathcal{X})$. Additionally, we may embed $\mathcal{P}^n(\mathcal{X})$ into $\mathcal{H}_\mathcal{I}\mathcal{P}^n(\mathcal{X})$ by recursively defining a sequence of maps $f_n : \mathcal{P}^n(\mathcal{X}) \to \mathcal{H}_\mathcal{I}\mathcal{P}^n(\mathcal{X})$ as follows:

$$
\begin{cases}
f_0 = id \\
f_{n+1} = c \circ push(f_n) & \text{if } n \geq 0.
\end{cases}
\tag{18}
$$

That is, $f_0$ is the identity map on $\mathcal{P}^0(\mathcal{X}) = \mathcal{X} = \mathcal{H}_\mathcal{I}\mathcal{P}^0(\mathcal{X})$, and $f_{n+1}$ is defined by composing the map $push(f_n) : \mathcal{P}(\mathcal{P}^n(\mathcal{X}) \to \mathcal{P}(\mathcal{H}_\mathcal{I}\mathcal{P}^n(\mathcal{X}))$ and the map $c : \mathcal{P}(\mathcal{H}_\mathcal{I}\mathcal{P}^n(\mathcal{X})) \to \mathcal{H}_\mathcal{I}\mathcal{P}^{n+1}(\mathcal{X})$ sending each distribution $\mu \in \mathcal{P}(\mathcal{H}_\mathcal{I}\mathcal{P}^n(\mathcal{X}))$ to the constant function $c(\mu)(i) = \mu$ for all $i \in \mathcal{I}$ in $Hom(\mathcal{I}, \mathcal{P}(\mathcal{H}_\mathcal{I}\mathcal{P}^n(\mathcal{X}))) = \mathcal{H}_\mathcal{I}\mathcal{P}^{n+1}(\mathcal{X})$.

**Lemma 12.** *The maps defined in Equation 18 are homeomorphisms onto their images.*

Hence, there is an induced map $f_\infty : \mathcal{P}^\infty(\mathcal{X}) \to \mathcal{H}_\mathcal{I}\mathcal{P}^\infty(\mathcal{X})$, with $f_\infty([\mu]) = [f_{r_{[\mu]}}(\mu)]$.

**Lemma 13.** *The map $f_\infty$ is a continuous bijection onto its image.*

We may also pass from $\mathcal{I}^n \times \mathcal{H}_\mathcal{I}\mathcal{P}(\mathcal{X})$ to $\mathcal{P}^n(\mathcal{X})$, essentially by selecting points $(i_1, ..., i_n) \in \mathcal{I}^n$ and evaluating the sequences of distributions at these indices. Formally, the maps $g_n : \mathcal{I}^n \times \mathcal{H}_\mathcal{I}\mathcal{P}^n(\mathcal{X}) \to \mathcal{P}^n(\mathcal{X})$ involved are defined recursively as follows:

$$
\begin{cases}
g_0 = id \\
g_{n+1} = push(g_n) \circ \otimes \circ (\delta_{\otimes^n} \times ev) & \text{if } n \geq 0.
\end{cases}
\tag{19}
$$

Here $ev : \mathcal{I} \times \mathcal{H}_\mathcal{I}\mathcal{P}^{n+1}(\mathcal{X}) \to \mathcal{P}(\mathcal{H}_\mathcal{I}\mathcal{P}^n(\mathcal{X}))$ is the map sending $(i, f)$ to the distribution $f(i)$; $\delta_{\otimes^n} : \mathcal{I}^n \to \mathcal{P}(\mathcal{I}^n)$ is the map sending $(i_1, ..., i_n)$ to the product measure $\delta_{i_1} \otimes \cdots \otimes \delta_{i_n}$; and $\otimes : \mathcal{P}(\mathcal{I}^n) \times \mathcal{P}(\mathcal{H}_\mathcal{I}\mathcal{P}^n(\mathcal{X})) \to \mathcal{P}(\mathcal{I}^n \times \mathcal{H}_\mathcal{I}\mathcal{P}^n(\mathcal{X}))$ sends $(\mu, \nu)$ to the product measure $\mu \otimes \nu$. For example, here are the first three maps:

$$
\begin{cases}
g_0(x) = x & x \in \mathcal{X} \\
g_1(i, f) = f(i) & i \in \mathcal{I}, \ f \in Hom(\mathcal{I}, \mathcal{P}(\mathcal{X})) \\
g_2(i_1, i_2, f) = push(ev)(\delta_{i_1} \otimes f(i_2)) & i_1, i_2 \in \mathcal{I}, \ f \in Hom(\mathcal{I}, \mathcal{H}_\mathcal{I}\mathcal{P}(\mathcal{X})).
\end{cases}
\tag{20}
$$

So, for example, for any measurable set $E \subseteq \mathcal{P}(\mathcal{X})$, we have:

$$
g_2(i_1, i_2, f)(E) = f(i_2)\left(\{h \in \mathcal{H}_\mathcal{I}\mathcal{P}(\mathcal{X}) | h(i_1) \in E\}\right).
\tag{21}
$$

In the above, note that $f(i_2) \in \mathcal{P}(\mathcal{H}_\mathcal{I}\mathcal{P}(\mathcal{X}))$.

We may then define a map $g_\infty : \mathcal{I}^\infty \times \mathcal{H}_\mathcal{I}\mathcal{P}^\infty(\mathcal{X}) \to \mathcal{P}^\infty(\mathcal{X})$ with $g_\infty(i_1, ..., [f]) = [g_{r_{[f]}}(i_1, ..., i_{r_{[f]}}, f)]$.

**Lemma 14.** *The map $g_n \circ (id_n \times f_n) : \mathcal{I}^n \times \mathcal{P}^n(\mathcal{X}) \to \mathcal{P}^n(\mathcal{X})$ defined by $(i_1, ..., i_n, \mu) \mapsto g_n(i_1, ..., i_n, f_n(\mu))$ is equal to the projection map $\pi_{n+1} : \mathcal{I}^n \times \mathcal{P}^n(\mathcal{X}) \to \mathcal{P}^n(\mathcal{X})$ defined by $\pi_{n+1}(i_1, ..., i_n, \mu) = \mu$.*

**Example: Multiple rounds of rock-paper-scissors.** Suppose now that Alice and Bob are playing multiple rounds of rock-paper-scissors, with games indexed by a parameter space $\mathcal{T}$, (for example, $\mathcal{T} = \{1, 2, 3, 4, ..., N\}$). Then the space of Bob's changing beliefs regarding Alice's gameplay is given by $\mathcal{H}_{\mathcal{T}}\mathcal{P}(\mathcal{X}) = Hom(\mathcal{T}, \mathcal{P}(\mathcal{X}))$. However, a sophisticated Bob will also be updating based upon his own uncertainty about those beliefs; this implies that Bob is operating in the space $\mathcal{H}_{\mathcal{T}}\mathcal{P}^2(\mathcal{X}) = Hom(T, \mathcal{P}(\mathcal{H}_{\mathcal{T}}\mathcal{P}(\mathcal{X})))$. Inductively, a Bob who incorporates information about uncertainty about the sequence of evolving beliefs in $\mathcal{H}_{\mathcal{T}}\mathcal{P}^n(\mathcal{X})$ is operating in the space $\mathcal{H}_{\mathcal{T}}\mathcal{P}^{n+1}(\mathcal{X}) = Hom(\mathcal{T}, \mathcal{P}(\mathcal{H}_{\mathcal{T}}\mathcal{P}^n(\mathcal{X})))$. Thus the analysis of all possible Bobs requires studying the colimit space $\mathcal{H}_{\mathcal{T}}\mathcal{P}^\infty(\mathcal{X})$. Players who play multiple rounds of rock-paper-scissors without updating their beliefs based upon gameplay may still be included in the analysis, as the space $\mathcal{P}^\infty(\mathcal{X})$ embeds bijectively into $\mathcal{H}_{\mathcal{T}}\mathcal{P}^\infty(\mathcal{X})$, as shown in Lemma 13.

## 3 EXAMPLES

**Single-point sets.** Suppose that $\mathcal{X} = \{*\}$ consists of a single point. Then $\mathcal{P}(\mathcal{X}) = \mathcal{P}_p(\mathcal{X})$ consists of a single point, as does $Hom(\mathcal{I}, \mathcal{P}(\mathcal{X}))$, since the only map is the map sending every element of $\mathcal{I}$ to the unique element of $\mathcal{P}(\mathcal{X})$. Hence, in all cases, the colimit spaces $\mathcal{P}^\infty(\mathcal{X})$, $\mathcal{P}_p^\infty(\mathcal{X})$, and $\mathcal{H}_{\mathcal{I}}\mathcal{P}^\infty(\mathcal{X})$ also consist of a single point.

**Trivial topology.** Let $\mathcal{X}$ be a non-empty topological space endowed with the trivial topology, i.e. the only open subsets of $\mathcal{X}$ are the empty set and $\mathcal{X}$ itself. There is a unique Borel probability distribution on $\mathcal{X}$, namely the one that assigns a probability of zero to the empty set and a probability of one to $\mathcal{X}$. Therefore, $\mathcal{P}(\mathcal{X}) = \mathcal{P}_p(\mathcal{X}) = \{*\}$ and $Hom(\mathcal{I}, \mathcal{P}(\mathcal{X}))$ consists of a single point. Therefore, the colimit spaces $\mathcal{P}^\infty(\mathcal{X})$, $\mathcal{P}_p^\infty(\mathcal{X})$, and $\mathcal{H}_{\mathcal{I}}\mathcal{P}^\infty(\mathcal{X})$ again consist of a single point.

**Competitive/Cooperative games.** Consider a competitive or cooperative game in which agents are reasoning about each other's strategies. For example, for two agents reasoning about each other's beliefs in discrete rounds of play, Agent 1 (A1) will select a particular point in the underlying space $\mathcal{X}$. Their beliefs about what state is correct in any given round may be modeled on the space $\mathcal{H}_{\mathbb{N}}\mathcal{P}(\mathcal{X}) = Hom(\mathbb{N}, \mathcal{P}(\mathcal{X}))$, where here we are endowing $\mathbb{N}$ with the discrete topology.[8] Agent 2 (A2) attempts to understand the beliefs of A1, and correspondingly has their own time-dependent beliefs in $\mathcal{H}_{\mathbb{N}}\mathcal{P}^2(\mathcal{X}) = Hom(\mathbb{N}, \mathcal{P}(\mathcal{H}_{\mathbb{N}}\mathcal{P}(\mathcal{X}))$. A1 reasons about what A2 is reasoning, and hence their beliefs over time may be modeled as an element of the space $\mathcal{H}_{\mathbb{N}}\mathcal{P}^3(\mathcal{X}) = Hom(\mathbb{N}, \mathcal{P}(\mathcal{H}_{\mathbb{N}}\mathcal{P}^2(\mathcal{X}))$. This process continues, as both agents attempt to reason about how the other agent is reasoning. In this sense, both agents' beliefs are modeled by elements of the space $\mathcal{H}_{\mathbb{N}}\mathcal{P}^\infty(\mathcal{X})$, and as soon as an agent decides their sequence of beliefs in some space $\mathcal{H}_{\mathbb{N}}\mathcal{P}^n(\mathcal{X}) = Hom(\mathbb{N}, \mathcal{P}(\mathcal{H}_{\mathbb{N}}\mathcal{P}^{n-1}(\mathcal{X}))$, then at every higher level of reasoning there is certainty: the agent knows what their beliefs at level $n$ are, and if this information is available to the other agent, then they can conclude what the other agent believes. Similarly, one could replace $\mathbb{N}$ with $\mathbb{R}$ for continuous-time reasoning.

**Hierarchical models.** Statistical models incorporating hierarchies may be modeled using the colimit structures as well. For example, in Bayesian hierarchical modeling (Gelman et al., 2004), one considers distributions on the hyperparameters, *hyperpriors*. Let $\mathcal{X}$ contain the possible data. Let $\Phi$ be a space of hyperparameters. Then there is a map $par : \Phi \to \mathcal{P}(\mathcal{X})$ defined by $par(\phi) = \mu_\phi$, the distribution parametrized by $\phi \in \Phi$. "Stage II" hyperpriors correspond to distributions in $\mathcal{P}(\Phi)$; however, these can also be thought of a distributions in $\mathcal{P}^2(\mathcal{X})$ in the following way: Consider the map $push(par) : \mathcal{P}(\Phi) \to \mathcal{P}^2(\mathcal{X})$. Then for any $E \subseteq \mathcal{P}(\mathcal{X})$ and $\psi \in \mathcal{P}(\Phi)$, we have the following:

$$push(par)(\psi)(E) = \psi\left(par^{-1}(E)\right) = \psi\left(\{\phi \in \Phi | \mu_\phi \in E\}\right). \tag{22}$$

Similarly, any "Stage n" hyperpriors corresponding to distributions in $\mathcal{P}^{n-1}(\Phi)$ give rise to distributions in $\mathcal{P}^n(\mathcal{X})$. Collecting all levels together gives rise to a map $par_\infty : \mathcal{P}^\infty(\Phi) \to \mathcal{P}^\infty(\mathcal{X})$ defined

---

[7]Recall that the discrete topology on a space $\mathcal{I}$ means that every subset of $\mathcal{I}$ is open.

[8]Other topologies are possible, and have different interpretations. In the other extreme, if $\mathbb{N}$ is endowed with the trivial topology, there is a unique Borel distribution on $\mathbb{N}$, and sampling randomly what the agents believe at any given time is more difficult.

by $par_\infty([\psi]) = [push^{r[\psi]}(par)(\psi)]$, where $push^k(par) = push \circ push \circ \cdots \circ push(par)$, where we are composing the push-forward maps $k$ times. In this way, the colimit $\mathcal{P}^\infty(\mathcal{X})$ contains all of the information about the hyperparameters and hyperpriors.

## 4 COMPUTING DISTANCE IN THE COLIMIT: ROCK-PAPER-SCISSORS

In order to quantify the distance between to different levels of Bob's reasoning, $\mu$ at level $M$ and $\nu$ at level $N$, we may compute $W_p^\infty([\mu], [\nu])$, where $\mu \in \mathcal{P}_p^M(\mathcal{X})$, $\nu \in \mathcal{P}_p^N(\mathcal{X})$, and $\mathcal{X} = \{\text{rock, paper, scissors}\}$. For simplicity, we consider level $M$ reasoning with level $N = 1$. For example, $\nu = \frac{1}{3}(\delta_{\text{rock}} + \delta_{\text{paper}} + \delta_{\text{scissors}})$ is a distribution in level $N = 1$. The computation of $W_p^\infty([\mu], [\nu])$ simplifies in this case as follows:

$$W_p^\infty([\mu], [\nu]) = \left( \inf_{\gamma^M \in \Gamma(\mu, \nu_M)} \int_{\mathcal{P}_p^{M-1}(\mathcal{X})} W_p^{M-1}(\alpha_{M-1}, \nu_{M-1})^p \, d\gamma^M(\alpha_{M-1}, \nu_{M-1}) \right)^{\frac{1}{p}}. \quad (23)$$

Here we have relied upon the notion of disintegration in order to simplify from integrating over $\mathcal{P}_p^{M-1}(\mathcal{X}) \times \mathcal{P}_p^{M-1}(\mathcal{X})$ to just integrating over $\mathcal{P}_p^{M-1}(\mathcal{X})$. We may repeat this process inductively until we have:

$$W_p^\infty([\mu], [\nu]) = \left( \inf_{\gamma^M \in \Gamma(\mu, \nu_M)} \int_{\mathcal{P}_p^{M-1}(\mathcal{X})} \inf_{\gamma^{M-1} \in \Gamma(\alpha_{M-1}, \nu_{M-1})} \int_{\mathcal{P}^{M-2}(\mathcal{X})} \cdots \right.$$

$$\left. \int_{\mathcal{P}_p(\mathcal{X})} \inf_{\gamma^1 \in \Gamma(\alpha_1, \nu)} \int_{\mathcal{X} \times \mathcal{X}} d(x, y) d\gamma^1(x, y) \cdots d\gamma^{M-1}(\alpha_{M-2}, \nu_{M-2}) d\gamma^M(\alpha_{M-1}, \nu_{M-1}) \right)^{\frac{1}{p}}. \quad (24)$$

Because $\mathcal{X}$ is finite in this case, the innermost integral becomes a finite sum, and finding the innermost infimum becomes equivalent to solving the following linear programming problem:

Let $\gamma = \gamma^1$ be a $\mathcal{X} \times \mathcal{X}$, here $3 \times 3$, matrix with non-negative entries. Assume that the distance metric on $\mathcal{X}$ is the discrete metric, i.e. $d(x, y) = 1$ if $x \neq y$ and 0 otherwise. We then wish to minimize $1 - \text{trace}(\gamma)$ subject to $\gamma \mathbf{1} = \alpha_\mathbf{1}$ and $\gamma^T \mathbf{1} = \nu$.

This optimization problem is readily solved, yielding $W_p^1(\alpha_1, \nu) = (|\alpha_1(\text{rock}) - 1/3| + |\alpha_1(\text{paper}) - 1/3|)^{1/p}$. As well, because $\nu_K$ is a Dirac distribution for each level, the infima may be removed, yielding:

$$W_p^\infty([\mu], [\nu]) =$$

$$\left( \int_{\mathcal{P}_p^{M-1}(\mathcal{X})} \int_{\mathcal{P}_p^{M-2}(\mathcal{X})} \cdots \int_{\mathcal{P}_p(\mathcal{X})} W_p^1(\alpha_1, \nu)^p d\alpha_2(\alpha_1) d\alpha_3(\alpha_2) \cdots d\alpha_{M-1}(\alpha_{M-2}) d\mu(\alpha_{M-1}) \right)^{1/p}$$

$$(25)$$

The measure $\mu$ exerts influence on these $\alpha$s through the outermost integral. The distance at level $M - 1$ is then the result of taking the expected infimum plan $\gamma$ at the level below and integrating to form the target $\alpha$ for the next level up. At every level above level 1, $\alpha$ will no longer be finite. The integrals can be approximated using the same strategy, where we consider minimizing the discretized approximation. Because in our example $\nu$ is a point at level 1, it remains a point at each higher level.

## 5 RELATED WORK

Colimits arise in category theory; category theory has been applied to problems in machine learning and inference. For example, Bayesian inference has been generalized to the study of particular types of categories in (Culbertson & Sturtz, 2013), though the context and goals of that paper take place within the framework of replacing statistical objects by their categorifications (Baez & Dolan, 1998). Gradient-based learning, probability, and equivariant learning have all been modeled using categorical methods (Schiebler et al., 2021), though the theory of colimits has not been extensively applied.

Metabeliefs appear in reinforcement learning, hierarchical/deep learning, and theory-of-mind reasoning. Reinforcement learning studies reasoning about the future and typically heuristically truncates reasoning by a horizon or similar mechanism. Other than proofs of asymptotic convergence, which do not reveal geometric and topological structure, we are not aware of efforts to understand limiting structures. Neural networks and Gaussian Processes of increasing depth have been studied, revealing pathologies and ways of understanding (Duvenaud et al., 2014; Peluchetti & Favaro, 2020; Sonoda & Murata, 2019; Roberts et al., 2022), but no comparable constructions of colimits that are as broad or as systematic. Recursive reasoning appears in theory of mind across economics (Brown, 1951; Camerer et al., 2004; Wright & Leyton-Brown, 2014), cognitive science (Wimmer & Perner, 1983; Hedden & Zhang, 2002; Wang et al., 2020), and machine learning (Heinrich & Silver, 2016; Baker et al., 2011; 2017; Wang et al., 2020; Shum et al., 2019), with geometric results for cooperative reasoning (Wang et al., 2019), but to our knowledge no systematic efforts toward generic construction or unification of uncertain reasoning.

## 6 CONCLUSION

Meta beliefs appear throughout machine learning when reasoning about the future, one's own, or others' beliefs. Using tools from category theory, we have analyzed the geometric and topological structure of infinite sequences of metabeliefs through the colimit space. Examples apply methods to understanding infinite hierarchies of beliefs as found in hierarchical and deep models, infinite recursions as in cooperative and competitive games, and Section 1.4 constructs spaces that apply to temporal structure. Moreover, we have shown how all of these problems are related in Section 2, thus offering a unified view of metabeliefs. Important future directions involve explicitly representing and understanding the metabelief structure into a broad set of machine learning models.

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
