## A  APPENDIX

This appendix contains the proofs of all lemmas.

*Lemma 2: Every equivalence class in $\mathcal{P}^\infty(\mathcal{X})$ is of the form $\{\mu, \delta^n(\mu), \delta^{n+1} \circ \delta^n(\mu), \delta^{n+2} \circ \delta^{n+1} \circ \delta^n(\mu), ...\}$ where $\mu$ is a unique distribution in $\mathcal{P}^n(\mathcal{X})$ such that $\mu$ is not a single-atom Dirac distribution.*

*Proof.* This follows directly from the definition of the equivalence relation, since for $\mu \in \mathcal{P}^i(\mathcal{X})$ and $\nu \in \mathcal{P}^j(\mathcal{X})$ such that there exists $k \in \mathbb{N}$, $i, j \leq k$ with $\delta^{ik}(\mu) = \delta^{jk}(\nu)$, we have by definition that $\delta^{k-1} \circ \cdots \circ \delta^i(\mu) = \delta^{k-1} \circ \cdots \circ \delta^j(\nu)$ are single-atom Dirac distributions, and two single-atom Dirac distributions are identical if and only if they have the same atom. That is, either $\nu = \delta^{ij}(\mu)$ if $i \leq j$ or $\mu = \delta^{ji}(\nu)$ if $j \leq i$. $\qquad\square$

*Lemma 3: Suppose that $\mathcal{X}$ is a compact T1 space [9] consisting of at least two distinct points. Then $\mathcal{P}^\infty(\mathcal{X})$ is non-compact with respect to the final topology.*

*Proof.* For each $n \geq 1$, fix some $\mu_n \in \mathcal{P}^n(\mathcal{X})$ such that $\mu_n$ is not a single-atom Dirac distribution. Let $P = \{[\mu_n]\}_{n \geq 1}$ be the subset of equivalence classes of these distributions in $\mathcal{P}^\infty(\mathcal{X})$. Then for any subset $Q \subseteq P$ and any $n \geq 1$, $(\Delta^n)^{-1}(Q) \cap \mathcal{P}^n(\mathcal{X})$ consists of at most one point. Because $\mathcal{X}$ is T1, each $\mathcal{P}^m(\mathcal{X})$, $m \in \mathbb{N}$ is T1, hence $(\Delta^n)^{-1}(Q) \cap \mathcal{P}^n(\mathcal{X})$ is closed. Therefore, since $\mathcal{P}^\infty(\mathcal{X})$ is the colimit of the sequence of $\mathcal{P}^n(\mathcal{X})$ spaces, every subset of $P$ is closed. Therefore, $\mathcal{P}^\infty(\mathcal{X})$ cannot be compact. $\qquad\square$

*Lemma 5: If $\mathcal{X}$ is Hausdorff, then so is $\mathcal{P}(\mathcal{X})$.*

*Proof.* Suppose that $\mu, \nu \in \mathcal{P}(\mathcal{X})$ such that $\mu \neq \nu$. Then there exists some Borel set $E \subseteq \mathcal{X}$ such that $\mu(E) \neq \nu(E)$. Without loss of generality, suppose $\mu(E) < \nu(E)$. Let $a \in \mathbb{R}$ such that $\mu(E) < a < \nu(E)$. Consider the characteristic function of $E$, $\mathbf{1}_E$, defined as $\mathbf{1}_E(x) = 1$ if $x \in E$ and $\mathbf{1}_E(x) = 0$ otherwise. Then observe that the following two sets are disjoint open sets with respect to the weak topology on $\mathcal{P}(\mathcal{X})$:

1. $E_{<a} := \{\eta \in \mathcal{P}(\mathcal{X}) | \eta(E) < a\}$

2. $E_{>a} := \{\eta \in \mathcal{P}(\mathcal{X}) | \eta(E) > a\}$

Then $\mu \in E_{<a}$ and $\nu \in E_{>a}$ and $E_{<a} \cap E_{>a} = \emptyset$. $\qquad\square$

*Lemma 6: $W_p^\infty$ is a metric on $\mathcal{P}_p^\infty(\mathcal{X})$*

*Proof.* Let $\mu \in \mathcal{P}_p^m(\mathcal{X})$ and $\nu \in \mathcal{P}_p^n(\mathcal{X})$ with $m \geq n$ such that $\mu$ and $\nu$ are not single-atom Dirac distributions (so they could be either different distributions or points in $\mathcal{X}$). By definition, $W_p^\infty([\mu], [\nu]) = W_p^m(\mu, \nu_m)$, where $\nu_m$ is the unique element in the intersection $[\nu] \cap \mathcal{P}_p^m$. Then because $W_p^m$ is a metric, $W_p^m(\mu, \nu_m) \geq 0$ with $W_p^m(\mu, \nu_m) = 0$ if and only if $\mu = \nu_m$. However, if $\mu = \nu_m$, then because we have chosen $m$ so that $\mu$ is not a single-atom Dirac distribution, $\nu_m$ is not a single-atom Dirac distribution. Therefore since $\nu_m \in [\nu]$ and $[\nu]$ is the unique non-single-atom Dirac distribution (or point in $\mathcal{X}$) in $[\nu]$, we have $\mu = \nu_m = \nu$. Similarly, $W_p^\infty([\mu], [\nu]) = W_p^\infty([\nu], [\mu])$ because $W_p^n(\mu, \nu) = W_p^n(\nu, \mu)$ for every $n \in \mathbb{N}$. All that remains is to verify the triangle inequality. Let $[\mu], [\nu], [\eta] \in \mathcal{P}_p^\infty(\mathcal{X})$ such that $\mu \in [\mu] \cap \mathcal{P}_p^{r_{[\mu]}}(\mathcal{X})$, $\nu \in [\nu] \cap \mathcal{P}_p^{r_{[\nu]}}(\mathcal{X})$, and $\eta \in [\eta] \cap \mathcal{P}_p^{r_{[\eta]}}(\mathcal{X})$. Without loss of generality, suppose that $r_{[\mu]} \leq r_{[\eta]}$. Then we have the following three cases:

1. $r_{[\nu]} \leq r_{[\mu]} \leq r_{[\eta]}$: In this case, we have the following:
$$W_p^\infty([\mu], [\nu]) + W_p^\infty([\nu], [\eta]) = W_p^{r_{[\mu]}}(\mu, \nu_{r_{[\mu]}}) + W_p^{r_{[\eta]}}(\nu_{r_{[\eta]}}, \eta) \qquad (26)$$
$$= W_p^{r_{[\eta]}}(\mu_{r_{[\eta]}}, \nu_{r_{[\eta]}}) + W_p^{r_{[\eta]}}(\nu_{r_{[\eta]}}, \eta) \geq W_p^{r_{[\eta]}}(\mu_{r_{[\eta]}}, \eta) = W_p^\infty([\mu], [\eta])$$

---

[9]This is equivalent to saying that for every $x \in \mathcal{X}$, the set $\{x\}$ is a closed subset of $\mathcal{X}$.

2. $r_{[\mu]} \leq r_{[\nu]} \leq r_{[\eta]}$: In this case, we have:

$$W_p^\infty([\mu], [\nu]) + W_p^\infty([\nu], [\eta]) = W_p^{r_{[\nu]}}(\mu_{r_{[\nu]}}, \nu) + W_p^{r_{[\eta]}}(\nu_{r_{[\eta]}}, \eta) \tag{27}$$

$$= W_p^{r_{[\eta]}}(\mu_{r_{[\eta]}}, \nu_{r_{[\eta]}}) + W_p^{r_{[\eta]}}(\nu_{r_{[\eta]}}, \eta) \geq W_p^{r_{[\eta]}}(\mu_{r_{[\eta]}}, \eta) = W_p^\infty([\mu], [\eta])$$

3. $r_{[\mu]} \leq r_{[\eta]} \leq r_{[\nu]}$: In this case, we have:

$$W_p^\infty([\mu], [\nu]) + W_p^\infty([\nu], [\eta]) = W_p^{r_{[\nu]}}(\mu_{r_{[\nu]}}, \nu) + W_p^{r_{[\nu]}}(\nu, \eta_{r_{[\nu]}}) \tag{28}$$

$$\geq W_p^{r_{[\nu]}}(\mu_{r_{[\nu]}}, \eta_{r_{[\nu]}}) = W_p^\infty([\mu], [\eta])$$

$\square$

*Lemma 8: Any space $\mathcal{X}$ may be embedded continuously in $\mathcal{H}_{\mathcal{I}}\mathcal{P}(\mathcal{X})$ via the following map, for any $x \in \mathcal{X}$ and $i \in \mathcal{I}$:*

$$cd(x)(i) := \delta_x. \tag{29}$$

*That is, $cd$ is the composition of the map sending each point $x$ to the single-atom Dirac distribution centered at $x$ and the constant-valued map sending each point in $\mathcal{I}$ to the aforementioned distribution.*

*Proof.* Let $E \subseteq \mathcal{H}_{\mathcal{I}}\mathcal{P}(\mathcal{X})$ open with respect to the compact-open topology. That is, $E = \bigcup_{a \in A} \left( \bigcap_{t_a=1}^{n_a} V(K_{t_a}, U_{t_a}) \right)$ for some indexing set $A$ (possibly uncountable), positive natural numbers $n_a$, compact subsets $K_{t_a} \subseteq \mathcal{I}$, and open subsets $U_{t_a} \subseteq \mathcal{P}(\mathcal{X})$. We must see if the set $cd^{-1}(E)$ is open in $\mathcal{X}$.

$$cd^{-1}(E) = \left\{ x \in \mathcal{X} \middle| c_{\delta_x} \in \bigcup_{a \in A} \left( \bigcap_{t_a=1}^{n_a} V(K_{t_a}, U_{t_a}) \right) \right\} \tag{30}$$

$$= \bigcup_{a \in A} \left\{ x \in \mathcal{X} \middle| c_{\delta_x} \in \bigcap_{t_a=1}^{n_a} V(K_{t_a}, U_{t_a}) \right\} = \bigcup_{a \in A} \left\{ x \in \mathcal{X} \middle| \delta_x \in \bigcap_{t_a=1}^{n_a} U_{t_a} \right\}$$

$$= \bigcup_{a \in A} \delta^{-1} \left( \bigcap_{t_a=1}^{n_a} U_{t_a} \right)$$

Note that $\bigcap_{t_a=1}^{n_a} U_{t_a}$ is a finite intersection of open sets and hence open, so because $\delta : \mathcal{X} \to \mathcal{P}(\mathcal{X})$ is continuous, $\delta^{-1}\left( \bigcap_{t_a=1}^{n_a} U_{t_a} \right)$ is open, and hence so is the final set in Equation 30. $\square$

*Lemma 9: Suppose that $\mathcal{X}$ is Hausdorff. Then each map $(cd)^n$ defined above is a homeomorphism onto its image.*

*Proof.* It is sufficient to show that that map $c : \mathcal{P}(\mathcal{X}) \to \mathcal{H}_{\mathcal{I}}\mathcal{P}(\mathcal{X})$ sending each distribution in $\mathcal{P}(\mathcal{X})$ to the constant function $c_\mu$ with $c_\mu(i) = \mu$ for every $i \in \mathcal{I}$, is a homeomorphism onto its image, because $cd : \mathcal{X} \to \mathcal{H}_{\mathcal{I}}\mathcal{P}(\mathcal{X})$ may then be written as the product of maps which are homeomorphisms onto their images; namely, $cd(x) = c \circ \delta(x)$. To this end, first note that the map $c$ is well-defined, since for each $\mu \in \mathcal{P}(\mathcal{X})$, $c_\mu$ is a continuous function. Furthermore, $c$ is injective, since for any $\mu, \nu \in \mathcal{P}(\mathcal{X})$, if $c_\mu = c_\nu$, then for every $i \in \mathcal{I}$, we have that $\mu = c_\mu(i) = c_\nu(i) = \nu$. Clearly $c$ is surjective onto its image. It remains to show that $c^{-1} : c(\mathcal{P}(\mathcal{X})) \to \mathcal{P}(\mathcal{X})$ is continuous. Suppose that $E \subseteq \mathcal{P}(\mathcal{X})$ is an open set with respect to the weak topology. We would like to show that $c(E)$ is open with respect to the subspace topology on $c(\mathcal{P}(\mathcal{X}))$. Since $\mathcal{X}$ is Hausdorff, $\mathcal{P}(\mathcal{X})$ is Hausdorff, and there exist disjoint open neighborhoods $N_\mu$ for each $\mu \in E$. Let $K \subseteq \mathcal{I}$ be an arbitrary non-empty compact subset. Then $\bigcup_{\mu \in E} V(K, N_\mu) \subseteq \mathcal{H}_{\mathcal{I}}\mathcal{P}(\mathcal{X})$ is open and $c(E) = c\left( \mathcal{P}(\mathcal{X}) \cap \left( \bigcup_{\mu \in E} V(K, N_\mu) \right) \right)$, hence $c(E)$ is open with respect to the subspace topology. Therefore $c$ is a homeomorphism onto its image, and hence so is $cd$. The claim of the lemma then follows inductively by replacing $\mathcal{X}$ with $\mathcal{H}_{\mathcal{I}}\mathcal{P}(\mathcal{X})$. $\square$

*Lemma 10: The following diagram commutes:*

$$
\begin{array}{ccc}
\mathcal{P}(\mathcal{I}) \times \mathcal{H}_{\mathcal{I}}\mathcal{P}^n(\mathcal{X}) & \xrightarrow{id \times \delta^0} & \mathcal{P}(\mathcal{I}) \times \mathcal{P}(\mathcal{H}_{\mathcal{I}}\mathcal{P}^n(\mathcal{X})) \\
{\scriptstyle id \times (cd)^n} \downarrow & & \downarrow {\scriptstyle \delta^1 \circ \pi_2} \\
\mathcal{P}(\mathcal{I}) \times \mathcal{H}_{\mathcal{I}}\mathcal{P}^{n+1}(\mathcal{X}) & \xrightarrow{push} & \mathcal{P}^2(\mathcal{H}_{\mathcal{I}}\mathcal{P}^n(\mathcal{X}))
\end{array}
$$

*Here $\pi_2 : \mathcal{P}(\mathcal{I}) \times \mathcal{P}(\mathcal{H}_{\mathcal{I}}\mathcal{P}^n(\mathcal{X})) \to \mathcal{P}(\mathcal{H}_{\mathcal{I}}\mathcal{P}^n(\mathcal{X}))$ is projection onto the second coordinate; i.e. $\pi_2(\mu, \nu) = \nu$.*

*Proof.* Start with any $(\mu, f) \in \mathcal{P}(\mathcal{I}) \times \mathcal{H}_{\mathcal{I}}\mathcal{P}^n(\mathcal{X})$. Then we have:
$$
push \circ (id \times (cd)^n)(\mu, f) = push(\mu, c_{\delta_f}) = c_{\delta_f}^*(\mu) \tag{31}
$$
Note that for any measurable $E \subseteq \mathcal{P}(\mathcal{H}_{\mathcal{I}}\mathcal{P}^n(\mathcal{X}))$, we have that $c_{\delta_f}^*(\mu)(E) = \mu(c_{\delta_f}^{-1}(E))$. Since $c_{\delta_f}$ is the constant map sending every element in $\mathcal{I}$ to the single-atom Dirac distribution $\delta_f$, we have that $c_{\delta_f}^{-1}(E) = \mathcal{I}$ if $\delta_f \in E$ and $c_{\delta_f}^{-1}(E) = \emptyset$ otherwise. Therefore, $\mu(c_{\delta_f}^{-1}(E)) = 1$ if $\delta_f \in E$ and $\mu(c_{\delta_f}^{-1}(E)) = 0$ otherwise. That is, $push \circ (id \times (cd)^n)(\mu, f) = \delta_{\delta_f}$.

On the other hand, we have:
$$
(\delta^1 \circ \pi_2) \circ (id \times \delta^0)(\mu, f) = \delta^1 \circ \pi_2(\mu, \delta_f) = \delta^1(\delta_f) = \delta_{\delta_f}. \tag{32}
$$
$\square$

Lemma 11: The following diagram commutes:

$$
\begin{array}{ccc}
\mathcal{I} \times \mathcal{H}_{\mathcal{I}}\mathcal{P}^n(\mathcal{X}) & \xrightarrow{\pi_2} & \mathcal{H}_{\mathcal{I}}\mathcal{P}^n(\mathcal{X}) \\
{\scriptstyle id \times (cd)^n} \downarrow & & \downarrow {\scriptstyle \delta} \\
\mathcal{I} \times \mathcal{H}_{\mathcal{I}}\mathcal{P}^{n+1}(\mathcal{X}) & \xrightarrow{ev} & \mathcal{P}(\mathcal{H}_{\mathcal{I}}\mathcal{P}^n(\mathcal{X}))
\end{array}
$$

*Proof.* This follows from Lemma 10 by taking only single-atom Dirac distributions in $\mathcal{P}(\mathcal{I})$; alternatively, one can check directly that for every $i \in \mathcal{I}$ and $h \in \mathcal{H}_{\mathcal{I}}\mathcal{P}^n(\mathcal{X})$, we have:
$$
\delta \circ \pi_2(i, h) = \delta(h) = \delta_h = c_{\delta_h}(i) = ev(i, cd^n(h)) = ev \circ (id \times (cd)^n)(i, h). \tag{33}
$$
$\square$

*Lemma 12: The maps defined in Equation 18 are homeomorphisms onto their images.*

*Proof.* We prove the claim by induction. Clearly $f_0$ is a homeomorphism onto its image. As well, we already know by Lemma 9 that the maps $c : \mathcal{P}(\mathcal{H}_{\mathcal{I}}\mathcal{P}^n(\mathcal{X})) \to \mathcal{H}_{\mathcal{I}}\mathcal{P}^{n+1}(\mathcal{X})$ are homeomorphisms onto their images. Now, assume that $f_n$ is a homeomorphism onto its image. It is well-known that the push-forward of a homeomorphism $f : \mathcal{A} \to \mathcal{B}$ is a homeomorphism $push(f) : \mathcal{P}(\mathcal{A}) \to \mathcal{P}(\mathcal{B})$ with respect to the weak topology for any Borel spaces $\mathcal{A}, \mathcal{B}$. Thus, $f_{n+1}$ is a composition of two homeomorphisms onto their images, and is therefore a homeomorphism onto its image. $\square$

*Lemma 13: The map $f_\infty$ is a continuous bijection onto its image.*

*Proof.* Let $\hat{f}_n : \mathcal{P}^n(\mathcal{X}) \to \mathcal{H}_\mathcal{I}\mathcal{P}^\infty(\mathcal{X})$ be the map $\mu \mapsto [f_n(\mu)]$. Then the following diagram commutes:

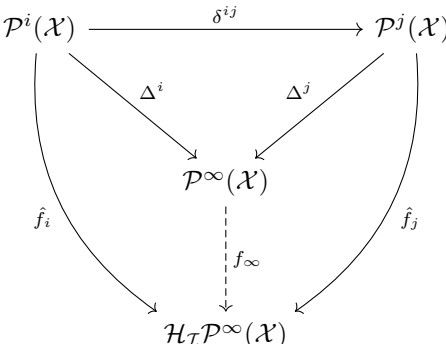

Thus, $f_\infty$ is a continuous bijective map. $\qquad\square$

*Lemma 14: The map $g_n \circ (id_n \times f_n) : \mathcal{I}^n \times \mathcal{P}^n(\mathcal{X}) \to \mathcal{P}^n(\mathcal{X})$ defined by $(i_1, ..., i_n, \mu) \mapsto g_n(i_1, ..., i_n, f_n(\mu))$ is equal to the projection map $\pi_{n+1} : \mathcal{I}^n \times \mathcal{P}^n(\mathcal{X}) \to \mathcal{P}^n(\mathcal{X})$ defined by $\pi_{n+1}(i_1, ..., i_n, \mu) = \mu$.*

*Proof.* We proceed by induction. For $n = 0$, we have $g_0 \circ (id_0 \times f_0) : \{\cdot\} \times \mathcal{X} \to \mathcal{X}$ is the identity map (identifying $\mathcal{I}^0 = \{\cdot\}$ with an arbitrary one-point set, and noting that $\mathcal{X}$ and $\{\cdot\} \times \mathcal{X}$ may be identified with each other). Hence, $g_0 \circ (id_0 \times f_0)(\cdot, x) = x$ for every $x \in \mathcal{X}$. Now, suppose that $g_n \circ (id_n \times f_n) = \pi_{n+1}$. Let $E \subseteq \mathcal{P}^n(\mathcal{X})$ be a Borel subset, and suppose $(i_1, ..., i_{n+1}) \in \mathcal{I}^{n+1}$ and $\mu \in \mathcal{P}^{n+1}(\mathcal{X})$. Then we have the following:

$$g_{n+1} \circ (id_{n+1} \times f_{n+1})(i_1, ..., i_{n+1}, \mu)(E) = g_{n+1}(i_1, ..., i_{n+1}, f_{n+1}(\mu))(E) \qquad (34)$$
$$= push(g_n)(\delta_{i_1} \otimes \cdots \otimes \delta_{i_n} \otimes push(f_n)(\mu))(E)$$
$$= (\delta_{i_1} \otimes \cdots \otimes \delta_{i_n} \otimes push(f_n)(\mu))\left(\{(j_1, ..., j_n, h) \in \mathcal{I}^n \times \mathcal{H}_\mathcal{I}\mathcal{P}^n(\mathcal{X}) | g_n(j_1, ..., j_n, h) \in E\}\right)$$
$$= push(f_n)(\mu)\left(\{h \in \mathcal{H}_\mathcal{I}\mathcal{P}^n(\mathcal{X}) | g_n(i_1, ..., i_n, h) \in E\}\right)$$
$$= \mu\left(\{\nu \in \mathcal{P}^n(\mathcal{X}) | g_n(i_1, ..., i_n, f_n(\nu)) \in E\}\right)$$
$$= \mu\left(\{\nu \in \mathcal{P}^n(\mathcal{X}) | (g_n \circ (id_n \times f_n)(i_1, ..., i_n, \nu) \in E\}\right)$$
$$= \mu\left(\{\nu \in \mathcal{P}^n(\mathcal{X}) | \pi_{n+1}(i_1, ..., i_n, \nu) \in E\}\right) = \mu(E)$$

$\qquad\square$