# OpenReview forum: "The (co)limit of metabeliefs"
_ICLR.cc/2024/Conference — Submitted to ICLR 2024_

### Official Review · Reviewer_BToD · 2023-10-30

**Soundness:** 3 good
**Presentation:** 2 fair
**Contribution:** 2 fair
**Rating:** 5
**Confidence:** 3

**Summary:**

The paper delves deep into the mathematical analysis of metabeliefs in machine learning. It adopts concepts from category theory, specifically colimits, to describe the structure and relationships of infinite sequences of metabeliefs. The primary objective is to offer a systematic and unified view of metabeliefs. The paper achieves this through a series of lemmas, definitions, and examples, particularly using the rock-paper-scissors game as a recurring motif.

**Strengths:**

**Originality:** The paper's approach to metabeliefs through category theory is innovative. The use of colimits provides a fresh perspective on the subject.

**Quality:** The mathematical derivations and lemmas presented are rigorous.

**Clarity:** While dense, the paper is consistent in its language and presentation.

**Significance:** The paper highlights the importance of understanding metabeliefs in machine learning, offering a theoretical foundation for further research.

**Weaknesses:**

**Accessibility:** The dense mathematical language can make it challenging for a broader audience to understand and appreciate the paper's contributions.

**Practical Application:** The paper leans heavily on theory, and more concrete examples or real-world applications might have bolstered its impact.

**Contextualization:** The paper could benefit from a clearer positioning within the broader landscape of machine learning research, specifically in terms of how it complements or diverges from existing works.

**Questions:**

1. Could the authors provide more concrete examples or real-world applications of their theory?

2. How does this work compare and contrast with other mathematical approaches to metabeliefs or similar constructs in machine learning?

3. Are there plans to test the presented theories empirically, or to develop algorithms/tools based on this framework?

---

> ### Author Response · Authors · 2023-11-17
> **Response to reviewer**
>
> We thank the reviewer for their comments.
>
> To respond to the questions:
> (1) Real-world implications of our theory, would be, for example, to choose the appropriate level of metabeliefs for specific problems, offer guarantees on approximation given a particular choice of level, and reason about compositions of problems.
> (2) We are unaware of any comparable effort to formulate the problem of metabeliefs. If the reviewer has specific suggestions, those would be appreciated.
> (3) Yes, we do plan to write subsequent papers on specific applications! As suggested in the introduction, we believe there are quite significant implications across learning, reinforcement learning, and social inference. In each case, engaging with the literate will require additional notion and technical machinery, and we therefore believe it best to tackle these applications in subsequent papers. If the reviewer raises their rating a bit, that would help us move faster toward that goal!
>
> Thanks again!

---

> > ### Comment · Reviewer_BToD · 2023-11-22
> >
> > Thanks for your response. I have a hard time connecting the theme of this paper to the scope of ICLR and my second question specifically mentioned "in machine learning".
> > After reading your response to other reviewers, I also was wondering how exactly "Beliefs are different from strategy" (in your response to Reviewer y6tA). In one sense, beliefs are our brain's strategy to maintain a world representation that has minimum inconsistencies and fits that of the social group around us.

---

> > > ### Author Response · Authors · 2023-11-23
> > >
> > > Yes, beliefs are different from strategy in that they formalize the problem of learning.

---

### Official Review · Reviewer_xqdq · 2023-11-01

**Soundness:** 3 good
**Presentation:** 4 excellent
**Contribution:** 2 fair
**Rating:** 5
**Confidence:** 2

**Summary:**

This paper explores the topological and geometrical structure of a sequence of beliefs, formalized as their colimit (under final topology) of the spaces of all levels of metabeliefs, which arises when modeling the reasoning about the future among agents in reinforcement learning, Bayesian hierarchical/deep learning, and game theory and economics.

This paper shows that commonly studies topological properties (such as metrizability, paracompactness, and Hausdorffness) carry over to the colimit of the directed systems pushed by both the weak topology (Theorem 4) or the induced topology of the Wasserstein $p$-distance (whose colimits have the same final topology, Theorem 7), and such constructions carry over to the index-dependent case (Section 2, which may formalize Bayesian hierarchical models with hyperpriors or iterated game such as the iterated rock-paper-scissors or iterated prisoner dilemma).

As a running example, the paper considers the colimit of the rock-paper-scissors game, and studies its geometrical properties (under Wasserstein $p$-distance, Section 4) in addition to its topological properties.

**Strengths:**

The categorical and topological exposition is very well written, for those familiar with such reasoning. The constructions (e.g., final topology on the colimit) are natural (in the categorical sense), and the results are somewhat intuitive and as expected.

**Weaknesses:**

The categorical and geometrical results, while natural, do not have foreseeable applications to machine learning (that I am aware of or can imagine). This is not unexpected for an exploratory paper to propose a new modeling perspective (colimit of topological spaces), but stronger connections to machine learning definitely help.

**Questions:**

While the authors argue that colimit of probability spaces (metrized under the Wasserstein $p$-distance) can model the geometry (and topology) of metabeliefs in co-operative and competitive games, are there examples showing that this modeling is better than other modeling for metabeliefs in terms _applications to machine learning?_

---

> ### Author Response · Authors · 2023-11-17
> **Response to reviewer**
>
> We thank the reviewer for their comments. To respond to the reviewer's question, we are not aware of any comparable methods of modeling metabeliefs to which we may compare. If the reviewer has specific examples in mind, those would be appreciated.
>
> We agree that understanding how the colimit might be useful in machine learning is important. We believe that applications are beyond the scope of the current paper, given the already significant technical content. To improve the clarity of the connection, we have elaborated the example in Section 4: Computing distance in the colimit: rock-paper-scissors. In particular, we give the solution to the $3\times 3$ optimization problem, and simplify the integration to remove the infima. We hope this helps make potential connections more clear.

---

> > ### Comment · Reviewer_xqdq · 2023-11-22
> >
> > Thank the authors for the response. However, I maintain my original rating and scores, as I would appreciate a more concrete connection to Machine Learning.

---

### Official Review · Reviewer_cWSP · 2023-11-03

**Soundness:** 3 good
**Presentation:** 3 good
**Contribution:** 2 fair
**Rating:** 5
**Confidence:** 1

**Summary:**

This paper studies chains of beliefs with tools from category theory. This is far outside my area of expertise.

**Strengths:**

N/A

**Weaknesses:**

I am not sure if the presentation is understand able to a wider audience, so I'm leaning negative with very low confidence.

**Questions:**

N/A

---

### Official Review · Reviewer_y6tA · 2023-11-04

**Soundness:** 2 fair
**Presentation:** 2 fair
**Contribution:** 2 fair
**Rating:** 3
**Confidence:** 4

**Summary:**

ICLR is not an appropriate venue for venue for this paper.


-------
Addendum:

Your last comment was "Great. It seems we agree that beliefs of finite depth are important." No, I'm claiming the opposite. Belief about belief about beliefs to finite depths are not useful and are misleading. The infinite limit (equilibrium) is useful, but as in the link I provided above, stopping at any finite depth often gives beliefs that are a function of the depth, not of anything else.  Beliefs about actions are imperative for intelligent action and AI, but that is not the same.

(I am not objecting to you claiming they are useful, I am objecting to you asserting they are useful without any evidence.  I'm also not clear why a category theory formalization is useful. Also, it isn't really relevant to ICLR.)

In another answer you said there wasn't a recent literature. Monte Carlo tree search (as, e.g., used in AlphaZero) can be seen as stochastic simulation through the space of belief about beliefs (where the beliefs are about the *actions* of the other agent); one reason it works is that it does not stop at an arbitrary point, but goes to the end of the episode/game. Also, (depending on what you mean by recent) Joe Halpern of Cornell has written many papers on this topic. A quick look found https://www.cs.cornell.edu/home/halpern/abstract.html#journal67

**Strengths:**

None that I can see.

**Weaknesses:**

The solution to the question at the end of the 4th paragraph after the abstract is "yes".  This problem was solved in 1950. The limit P^\inf(X) is the fixed point which is the definition of a Nash equilibrium. Nash proved the existence of a fixed point, and there is a considerable literature on how to compute them and its complexity (PPAD). The Nash equilibrium has a maximization step (agent agent has a utility it is trying to maximize), which seems essential to the running example, but doesn't  appear in the paper. (One of) the brilliance(s) of Nash's result is that an agent does not need to do the recursive reasoning that this paper is about. We can compute the fixed point directly.

The rest of the running example is all nonsense. "In order for Bob to generate his own strategy, he must consider what he knows..." is not true. Bob doesn't need to do this recursive reasoning.

**Questions:**

Am I wrong?

---

> ### Author Response · Authors · 2023-11-17
> **Response to your question, and thank you!**
>
> We wish to thank the reviewer for their question. To respond directly: The reviewer is incorrect, Nash equilibria are different from colimit spaces induced by metabeliefs. Specifically, the question at the end of paragraph 4 is ``Can we consider beliefs at all levels simultaneously?''. Beliefs are different from strategy. Nash equilibria assume away the problem of beliefs and this is one reason that such equilibria are not effective models of behavior. Our contribution is to offer a precise mathematical formalization of reasoning about metabeliefs of all levels.
>
> More technically:
> - The reviewer mentions that ``the limit $P^{\infty}(\mathcal{X})$ is the fixed point which is the definition of a Nash equilibrum...." The space $P^{\infty}(\mathcal{X})$ here is a topological space consisting of every distribution in $\mathcal{P}^n(\mathcal{X})$ for every finite $n$, as well as points of $\mathcal{X}$, modulo identifying the embedding of each $\mathcal{P}^n(\mathcal{X})$ into $\mathcal{P}^{n+1}(\mathcal{X})$. To say that this space is a Nash equilbrium does not make sense: the space is a collection of equivalence classes with a topology imposed on them, a Nash equilibrium is a set of strategies for each player in a non-cooperative game such that each player has no preference for switching strategies.
> - The setting of Nash equilibria applies to non-cooperative games in which each player knows the equilibria strategies of the other players. Our setting is applicable to non-cooperative games, cooperative games, and things that are not games, and is focused in the rock-paper-scissors example on Bob not knowing Alice's strategy, her equilibrium strategy, or even exactly what her space of strategies might be.
> - In order to fix the confusion surrounding the rock-paper-scissors example, we have changed "In order for Bob to generate his own strategy, he must consider what he knows...." from the example on page 4 to the following: "In order for Bob to generate his own strategy, he may consider what he knows...."
>
> We apologize for any confusion created by our wording, and thank the reviewer for their comments.

---

> ### Comment · Reviewer_y6tA · 2023-11-17
> **Thinking recursively to arbitrary depths about thinking does not work.**
>
> The point of that Nash equilibrium is that we don't need or want to do the infinite regress of meta-beliefs. It is not as though the people who have studied this over the decades did not explore the tree of meta-beliefs; it's just that it is too difficult and not needed,  "every distribution... for every finite n" is the wrong thing to do. Every finite set of beliefs about beliefs can give the wrong answer!! (Rock-paper-scissors is too symmetric to explain the problem; I found a good example at https://artint.info/3e/html/ArtInt3e.Ch14.S4.html). In such examples, every finite sequence of meta beliefs gives an arbitrary answer. It's the fixed point / equilibrium, not the space of all finite parts, that forms the rational belief. An agent doesn't need to do the reasoning about meta-beliefs to arbitrary depths, which you just assume is a useful thing to do. "Nash equilibria assume away the problem of beliefs" - precisely because the infinite regress is the wrong thing to do (and finite subparts are even more useless). In the paper "In practice, metabeliefs are typically truncated to a finite sequence." isn't true; it is typical that the equilibrium is used (discounting means  finite part is a good approximation to the equilibrium).
>
> In a Nash equilibria an agent doesn't need to know the strategies of the other players (perhaps selecting which Nash equilibrium, but is there is only one then there is no reason to deviate). There are related equilibria (such as correlated equilibria) for other games (where "game" is a very general concept of multi-agent interaction).  There is a reason they rejected the explicit (infinite) regress of meta-beliefs.
>
> The motivation of the paper seemed to be mis-guided. Perhaps there is a venue for this paper, but ICLR isn't it. (Perhaps KR or AAMAS).

---

> > ### Author Response · Authors · 2023-11-21
> > **Metabeliefs are important for modeling subjective beliefs**
> >
> > Note that our set-up models the goalkeeper scenario as well as scenarios in which one agent has more information than the other, that neither agent has perfect or complete information, that one or both agents are constrained, that there is only a single agent, or that the agent is reasoning about themself.
> >
> > Our approach also handles compositions of such possibilities. For example, suppose that the goalkeeper is aware of the provided joint distribution, but also knows that the kicker suffers from migraines when it is sunny limiting their ability to handle uncertainty, that the kicker is learning to kick, or that the opposition exhibits (a priori unknown) sequential dependencies in their play. All will change the joint distribution, making it no longer static, so the goalkeeper must have a probability distribution on the space of such joint distributions, depending upon the weather, learning and/or candidate dependencies.
> >
> > The reviewer seems focused on the specific case of simple economic games, for which there are equilibrium concepts. Our emphasis is on the general case of subjective decision making in which beliefs are relevant and critical (see e.g. Savage, 1954), which include simple games as special cases. We are not aware of any existing approach of comparable generality.

---

> ### Comment · Reviewer_y6tA · 2023-11-22
>
> I don't disagree that agents need to reason about the probability of the other agents actions. The other agent has beliefs that result in a distribution over their actions. It is that distribution over actions that an agent needs to find the best response to.  None of your examples seems to justify beliefs about beliefs to an arbitrary depth.
>
> There is a reason why there is not a recent literature on meta-beliefs to arbitrary depths. It doesn't work and we have much better (and simpler!) solutions.

---

> > ### Author Response · Authors · 2023-11-23
> >
> > Great. It seems we agree that beliefs of finite depth are important. One important contribution of our paper is providing a way of thinking about the choice of which finite depth. Again, we are unaware of approaches that are comparably general and simpler.

---

### Official Review · Reviewer_sas8 · 2023-11-09

**Soundness:** 3 good
**Presentation:** 4 excellent
**Contribution:** 3 good
**Rating:** 8
**Confidence:** 4

**Summary:**

This paper deals with the limit of beliefs of beliefs.
In a very general setting, without making many structural assumptions, the authors perform a theoretical study of the properties of the limiting space. Their first main theoretical contributions is a result guaranteeing that the limiting space is Hausdorff when the initial space is. They continue their investigation by focussing on distributions with finite p-moments (almost everywhere), which is where their second main theoretical distribution is: they identify and characterise the topology on the limiting space in this setting. Besides this, the paper shows connections between these two settings, ans well as relations with a setting of index-dependent distributions.
The authors end their paper with some examples of specific structures, with an summary of how to compute the distance in a simple 3-state game, and with an overview of related work.

**Strengths:**

To the best of my knowledge, the content and problem setting are original.

This paper is well written, and contains clear definitions and results.
I appreciate the structure of the paper.
Overall, the paper clearly shows that the authors thought well about their presentation.
I particularly appreciate the clear set up for each of the three settings; in particular the first and second one.

The main strength of this paper are the large number of theoretical results.
The authors manage to both clearly define all the objects, as well as derive many results in the main text, in a limited space, for which I want to commend them.
I have tried and checked the proofs.
To the best of my knowledge, they are correct—bar two questions I have later on with hopefully minor consequences.

I also like the philosophical justification for the problem under investigation. I admit that I am sceptical about many of the justifications for higher order beliefs, but the limiting approach that the authors take seem well founded, and alleviates some of my initial reservations. I like that the authors do not assume that there is a boundary of the level of uncertainty of uncertainty

In section 4, the example that shows the most promise in my opinion, is the one about hierarchical models. I like that the authors identify the relevant concepts in their setting.

**Weaknesses:**

I don't think the paper has major weaknesses.
Its focus certainly is on the theoretical front, so one might argue that the paper lacks applications, simulations, or computational details; however, given the large number of results I would tend to find this of minor importance here.

As I indicated above, the writing style is very clear. However, I found myself sometimes lost in Section 2, which contains abstract material. I believe that, given the page limitations, it is difficult to improve this, but perhaps some clarifications are possible.

I believe that there on two technical parts, there are details missing. I'll list the details in the section Questions underneath, but they concern Lemma 2 and Lemma 6. I am hopeful, however, that even if I am right, this won't have major consequences for the rest of the story and results.

**Questions:**

I start with my two main questions:

1/ page 3, Lemma 2: I think that the authors need to be careful with the range of n. More specifically, what happens if n=0? Does the metric space {\cal X} contain distributions?
- If not, can we always assume that n≥1? Why?
- If so, that is a structural assumption on {\cal X} that seems strange, and needs to be spelled out.
It could be that I misunderstand something here. What could improve my understanding, is having the answer to the following question: What is the form of the equivalence class that contains an element of {\cal X}?

2/ page 5, Lemma 6: The authors don't prove that the alleged metric is positive definite (that W(mu,nu)>0 if mu≠nu). I believe that this is a requirement of the usual definition of a metric. Do the authors need this? If so, does their map W satisfy this?

I conclude my review with a list of some small remarks and typos:

page 1, first paragraph following the abstract: "... to sequences of beliefs quantify uncertainty ..." should read something like "... to sequences of beliefs that quantify uncertainty ..." (so sets with a "that", or an alternative).

page 1, second paragraph: There is a dot missing after the third sentence.

page 1, fourth paragraph: "... one must play of the three options." should read "... one must play one of the three options.".

page 1, fifth paragraph: "geoemtric" should read "geometric". In the following sentence, the word "gives" is superfluous.

page 2, paragraph following Theorem 1: "Parts (2)--(6)" should be "Parts (2)--(5)", I believe, since there are only 5 parts in Theorem 1.

page 3, Lemma 3: In the proof, the authors use a result that guarantees that each {\cal P}^m({\cal X}) is T1 if {\cal X} is T1. While I can believe this is true, I would like to have a proof. Is this a standard
result? If so, it might be worth giving a reference.

page 3, Lemma 5: Its proof appears in the supplementary material as well as in the paper; perhaps the copy in the paper can be omitted?

page 4, proof of Theorem 4: Theorem 1 part (4) talks about *compact* spaces, and not about the weaker notion of *paracompact* spaces. Why can this be inferred? If that's not immediate, perhaps it suffices for the
further applications that the spaces are compact, in addition to being paracompact?

page 5, Lemma 6: There is a dot missing at the end of the lemma.

page 5, right hand side of Equation (10): the part of the expression after | is not a condition, which reads a bit strangely. I suggest that the authors add that \tilde\nu \in the sets shown, if that's correct.

page 5, Equation (11): Why is does \Delta^i commute with \bigcap and \bigcup? This must be some standard result, but I don't seem to find a reference for this.

page 6, Lemma 11: The authors should define the map ev already here, as they use it in their commuting diagram. Now they only define the map ev later on in Section 2.

page 8, subsection "Competitive/Cooperative games": There is a dot lacking between the first two sentences. Also, "In this sense, both agents beliefs ..." should read "In this sense, both agents' beliefs ...".

---

> ### Author Response · Authors · 2023-11-17
> **Thank you for the careful read and detailed comments!**
>
> We wish to thank the reviewer for their enthusiastic support and helpful and detailed feedback. We have addressed the reviewer's two main questions as follows:
>
> - On page 3 in Lemma 2, we have added the following clarification for $n=0$: In place of "...$\mu$ is a unique distribution in $\mathcal{P}^n(\mathcal{X})$ such that $\mu$ is not a single-atom Dirac distribution." We replace with: "...$\mu$ is a unique distribution in $\mathcal{P}^n(\mathcal{X})$ such that $\mu$ is not a single-atom Dirac distribution or $\mu$ is a unique point in $\mathcal{X}$ (recall that $\delta^0(x)=\delta_x$ is the Dirac distribution in $\mathcal{P}(\mathcal{X})$ with support equal to $\{x\}$)."
> - $W_p^{\infty}$ is indeed positive definite. We have added this proof to the proof of Lemma 6 given in the appendix: ``Let $\mu\in\mathcal{P}_p^m(\mathcal{X})$ and $\nu\in\mathcal{P}_p^n(\mathcal{X})$ with $m\geq n$ such that $\mu$ and $\nu$ are not single-atom Dirac distributions (so they could be either different distributions or points in $\mathcal{X}$). By definition, $W_p^{\infty}([\mu],[\nu])=W_p^m(\mu,\nu_m)$, where $\nu_m$ is the unique element in the intersection $[\nu]\cap\mathcal{P}_p^m$. Then because $W_p^m$ is a metric, $W_p^m(\mu,\nu_m)\geq 0$ with $W_p^m(\mu,\nu_m)=0$ if and only if $\mu=\nu_m$. However, if $\mu=\nu_m$, then because we have chosen $m$ so that $\mu$ is not a single-atom Dirac distribution, $\nu_m$ is not a single-atom Dirac distribution. Therefore since $\nu_m\in [\nu]$ and $[\nu]$ is the unique non-single-atom Dirac distribution (or point in $\mathcal{X}$) in $[\nu]$, we have $\mu=\nu_m=\nu$."
>
> We also fix all of the typos in the list provided, and wish to once again thank the reviewer for providing such a detailed list.
> Here are some more detailed responses to a few of the small remarks:
>
> - The T1 claim may be found in Parthasarathy (2005) "Introduction to Probability and Measure", though the statement can also be amended to change T1 to, say, locally compact and metrizable, and the statement is easy to prove directly.
> - Many of the applications (particularly to machine learning) implicitly use compact spaces; however in the statement of part 4 of Theorem 1, "compact" can be changed to "paracompact," and the statement is still true (albeit weaker). We have amended this.
> - We have fixed the right-hand side of Equation 10 by removing the unnecessary and confusing portions of the notation: the beginning and ending curly braces and the $\nu\in\mathcal{P}_p^i(\mathcal{X})$.
> - $\Delta^i$ commutes with $\bigcup$ since for any function $f$ and sets $A,B$, $f(A\cup B)=f(A)\cup f(B)$. To make the passage from equation 11 to equation 12 more clear, we have added a few lines to the proof (including why $\Delta^i$ commutes with $\bigcap$). This addition begins immediately following Equation 11, "Applying $\Delta^i$..." and continues until just before Equation 14, terminating with, "Now, Equation 13 becomes:".

---

### Meta-Review · Area_Chair_XjPb · 2023-12-11

**Metareview:**

This paper presents a theoretical study of beliefs of beliefs and properties in the limit. The paper is highly theoretical and arguably hard for the audience of the conference to follow, which makes me suggest that this is not a good place for its publication. The work has led to interesting discussions in the reviewing process, with clearly very different opinions by different committee members, while others were too far in terms of topic to provide valuable input for discussions. All in all, there is evidence that this is not the ideal venue for the paper, or at least it was not for the subsample consisting of the committee members involved in the evaluation. It is hoped that the authors can make use of the feedback from reviewers, which comes in different flavours, in order to continue their journey.

**Justification For Why Not Higher Score:**

Short to medium term impact and suitability for the conference are to be considered.

**Justification For Why Not Lower Score:**

N/A

---

### Decision · Program_Chairs · 2024-01-16

Reject